# On the Posterior Distribution in Denoising: Application to Uncertainty Quantification

**Hila Manor**
Faculty of Electrical and Computer Engineering
Technion – Israel Institute of Technology
`hila.manor@campus.technion.ac.il`

**Tomer Michaeli**
Faculty of Electrical and Computer Engineering
Technion – Israel Institute of Technology
`tomer.m@ee.technion.ac.il`

## Abstract

Denoisers play a central role in many applications, from noise suppression in low-grade imaging sensors, to empowering score-based generative models. The latter category of methods makes use of Tweedie's formula, which links the posterior mean in Gaussian denoising (*i.e.*, the minimum MSE denoiser) with the score of the data distribution. Here, we derive a fundamental relation between the higher-order central moments of the posterior distribution, and the higher-order derivatives of the posterior mean. We harness this result for uncertainty quantification of pre-trained denoisers. Particularly, we show how to efficiently compute the principal components of the posterior distribution for any desired region of an image, as well as to approximate the full marginal distribution along those (or any other) one-dimensional directions. Our method is fast and memory-efficient, as it does not explicitly compute or store the high-order moment tensors and it requires no training or fine tuning of the denoiser. Code and examples are available on the project [website](#).

## 1 Introduction

Denoisers serve as key ingredients in solving a wide range of tasks. Indeed, along with their traditional use for noise suppression (Aharon et al., 2006; Buades et al., 2005; Dabov et al., 2007; Krull et al., 2019; Liang et al., 2021; Portilla et al., 2003; Roth & Black, 2009; Rudin et al., 1992; Zhang et al., 2017a; 2021), the last decade has seen a steady increase in their use for solving other tasks. For example, the plug-and-play method (Venkatakrishnan et al., 2013) demonstrated how a denoiser can be used in an iterative manner to solve arbitrary inverse problems (*e.g.*, deblurring, inpainting). This approach was extended by many, and has led to state-of-the-art results on various restoration tasks (Brifman et al., 2016; Romano et al., 2017; Tirer & Giryes, 2018; Zhang et al., 2017b). Similarly, the denoising score-matching work (Vincent, 2011) showed how a denoiser can be used for constructing a generative model. This approach was later improved (Song & Ermon, 2019), and highly related ideas (originating from (Sohl-Dickstein et al., 2015)) served as the basis for diffusion models (Ho et al., 2020), which now achieve state-of-the-art results on image generation.

Many of the uses of denoisers rely on Tweedie's formula (often attributed to Robbins (1956), Miyasawa et al. (1961), Stein (1981), and Efron (2011)) which connects the MSE-optimal denoiser for white Gaussian noise, with the score function (the gradient of the log-probability w.r.t the observations) of the data distribution. The MSE-optimal denoiser corresponds to the posterior mean of the clean signal conditioned on the noisy signal. Therefore, Tweedie's formula in fact links between the first posterior moment and the score of the data. A similar relation holds between the second posterior moment (*i.e.*, the posterior covariance) and the second-order score (*i.e.*, the Hessian of the log-probability w.r.t the observations) (Gribonval, 2011), which is in turn associated with the derivative (*i.e.*, Jacobian) of the posterior moment. Recent works used this relation to quantify uncertainty in denoising (Meng et al., 2021), as well as to improve score-based generative models (Dockhorn et al., 2022; Lu et al., 2022; Meng et al., 2021; Mou et al., 2021; Sabanis & Zhang, 2019).

In this paper we derive a relation between higher-order posterior central moments and higher-order derivatives of the posterior mean in Gaussian denoising. Our result provides a simple mechanism that, given the MSE-optimal denoiser function and its derivatives at some input, allows determining

the entire posterior distribution of clean signals for that particular noisy input (under mild conditions). Additionally, we prove that a similar result holds for the posterior distribution of the projection of the denoised output onto a one-dimensional direction.

We leverage our results for uncertainty quantification in Gaussian denoising by employing a pre-trained denoiser. Specifically, we show how our results allow computing the top eigenvectors of the posterior covariance (*i.e.*, the posterior principal components) for any desired region of the image. We further use our results for approximating the entire posterior distribution along each posterior principal direction. As we show, this provides valuable information on the uncertainty in the restoration. Our approach uses only forward passes through the pre-trained denoiser and is thus advantageous over previous uncertainty quantification methods. In particular, it is training-free, fast, memory-efficient, and applicable to high-resolution images. We illustrate our approach with several pre-trained denoisers on multiple domains, showing its practical benefit in uncertainty visualization.

## 2 RELATED WORK

Many works studied theoretical properties of MSE-optimal denoisers for signals contaminated by additive white Gaussian noise. Perhaps the most well-known result is Tweedie's formula (Efron, 2011; Miyasawa et al., 1961; Robbins, 1956; Stein, 1981), which connects the MSE-optimal denoiser with the score function of noisy signals. Another interesting property, shown by Gribonval (2011), is that the MSE-optimal denoiser can be interpreted as a maximum-a-posteriori (MAP) estimator, but with a possibly different prior. The work most closely related to ours is that of Meng et al. (2021), who studied the estimation of high-order scores. Specifically, they derived a relation between the high-order posterior non-central moments in a Gaussian denoising task, and the high-order scores of the distribution of noisy signals. They discussed how these relations can be used for learning high-order scores of the data distribution. But due to the large memory cost of storing high-order moment tensors, and the associated computational cost during training and inference, they trained only second-order score models and only on small images (up to $32 \times 32$). They used these models for predicting the posterior covariance in denoising tasks, as well as for improving the mixing speed of Langevin dynamics sampling. Their result is based on a recursive relation, which they derived, between the high-order derivatives of the posterior mean and the high-order *non-central* moments of the posterior distribution in Gaussian denoising. Specifically, they showed that the non-central posterior moments $\boldsymbol{m}_1, \boldsymbol{m}_2, \boldsymbol{m}_3, \ldots$, admit a recursion of the form $\boldsymbol{m}_{k+1} = f(\boldsymbol{m}_k, \nabla \boldsymbol{m}_k, \boldsymbol{m}_1)$.

In many settings, *central* moments are rather preferred over their non-central counterparts. Indeed, they are more numerically stable and relate more intuitively to uncertainty quantification (being directly linked to variance, skewness, kurtosis, etc.). Unfortunately, the result of (Meng et al., 2021) does not trivially translate into a useful relation for central moments. Specifically, one could use the fact that the $k$th central moment, $\boldsymbol{\mu}_k$, can be expressed in terms of $\{\boldsymbol{m}_j\}_{j=1}^k$, and that each $\boldsymbol{m}_j$ can be written in terms of $\{\boldsymbol{\mu}_i\}_{i=1}^j$. But naively substituting these relations into the recursion of Meng et al. (2021) leads to an expression for $\boldsymbol{\mu}_k$ that includes all lower-order central-moments and their high-order derivatives. Here, we manage to prove a very simple recursive form for the central moments, which takes the form $\boldsymbol{\mu}_{k+1} = \tilde{f}(\boldsymbol{\mu}_k, \nabla \boldsymbol{\mu}_k, \boldsymbol{\mu}_2)$. Another key contribution, which we present beyond the framework studied by Meng et al. (2021), relates to marginal posterior distributions along arbitrary cross-sections. Specifically, we prove that the central posterior moments of any low-dimensional projection of the signal, also satisfy a similar recursion. Importantly, we show how these relations can serve as very powerful tools for uncertainty quantification in denoising tasks.

Uncertainty quantification has drawn significant attention in the context of image restoration. Many works focused on per-pixel uncertainty prediction (Angelopoulos et al., 2022; Gal & Ghahramani, 2016; Horwitz & Hoshen, 2022; Meng et al., 2021; Oala et al., 2020), which neglects correlations between the uncertainties of different pixels in the restored image. Recently, several works forayed into more meaningful notions of uncertainty, which allow to reason about semantic variations (Kutiel et al., 2023; Sankaranarayanan et al., 2022). For example, a concurrent work by Nehme et al. (2023) presented a method for learning the posterior principal components of arbitrary inverse problems. However, all existing methods either require a pre-trained generative model with a disentangled latent space (*e.g.*, StyleGAN (Karras et al., 2020) for face images) or, like many of their per-pixel

counterparts, require training. Here we present a training-free, computationally efficient, method that only requires access to a pre-trained denoiser.

# 3 MAIN THEORETICAL RESULT

We now present our main theoretical result, starting with scalar denoising and then extending the discussion to the multivariate setting. The scalar case serves two purposes. First, it provides intuition. But more importantly, the formulae for moments of orders higher than three are different for the univariate and multivariate settings, and therefore the two cases require separate treatment.

## 3.1 THE UNIVARIATE CASE

Consider the univariate denoising problem corresponding to the observation model
$$y = x + n, \tag{1}$$
where x is a scalar random variable with probability density function $p_x$ and the noise $n \sim \mathcal{N}(0, \sigma^2)$ is statistically independent of x. The goal in denoising is to provide a prediction $\hat{x}$ of x, which is a function of the measurement y. It is well known that the predictor minimizing the MSE, $\mathbb{E}[(x - \hat{x})^2]$, is the posterior mean of x given y. Specifically, given a particular measurement $y = y$, the MSE-optimal estimate is the first moment of the posterior density $p_{x|y}(\cdot|y)$, which we denote by
$$\mu_1(y) = \mathbb{E}[x|y = y]. \tag{2}$$

While optimal in the MSE sense, the posterior mean provides very partial knowledge on the possible values that x could take given that $y = y$. More information is encoded in higher-order moments of the posterior. For example, the posterior variance provides a measure of uncertainty about the MSE-optimal prediction, the posterior third moment provides knowledge about the skewness of the posterior distribution, and the posterior fourth moment can already reveal a bimodal behavior.

Let us denote the higher-order posterior central moments by
$$\mu_k(y) = \mathbb{E}\left[(x - \mu_1(y))^k \,\middle|\, y = y\right], \quad k \geq 2. \tag{3}$$
Our key result is that knowing the posterior mean function $\mu_1(\cdot)$ and its derivatives at $y$ can be used to recursively compute all higher-order posterior central moments at $y$ (see proof in App. A).

**Theorem 1** (Posterior moments in univariate denoising). *In the scalar denoising setting of (1), the high-order posterior central moments of* x *given* y *satisfy the recursion*
$$\mu_2(y) = \sigma^2 \mu_1'(y),$$
$$\mu_3(y) = \sigma^2 \mu_2'(y),$$
$$\mu_{k+1}(y) = \sigma^2 \mu_k'(y) + k\mu_{k-1}(y)\mu_2(y), \quad k \geq 3. \tag{4}$$
*Thus,* $\mu_{k+1}(y)$ *is uniquely determined by* $\mu_1(y), \mu_1'(y), \mu_1''(y), \ldots, \mu_1^{(k)}(y)$.

Figure 1 illustrates this result via a simple example. Here, the distribution of x is a mixture of two Gaussians. The left pane depicts the posterior density $p_{x|y}(\cdot|\cdot)$ as well as the posterior mean function $\mu_1(\cdot)$. We focus on the measurement $y = y^*$, shown as a vertical dashed line, for which the posterior $p_{x|y}(\cdot|y^*)$ is bimodal (right pane). This property cannot be deduced by merely examining the MSE-optimal estimate $\mu_1(y^*)$. However, this information does exist in the derivatives of $\mu_1(\cdot)$ at $y^*$. To demonstrate this, we numerically differentiated $\mu_1(\cdot)$ at $y^*$, used the first three derivatives to extract the first four posterior moments using Theorem 1, and computed the maximum entropy distribution that matches those moments (Botev & Kroese, 2011). As can be seen, this already provides a good approximation of the general shape of the posterior (dashed red line).

Theorem 1 has several immediate implications. First, it is well known that if the moments do not grow too fast, then they uniquely determine the underlying distribution (Lin, 2017). This is the case *e.g.*, for distributions with a compact support and is thus relevant to images, whose pixel values typically lie in $[0, 1]$. For such settings, Theorem 1 implies that knowing the posterior mean at the neighborhood of some point $y$, allows determining the entire posterior distribution for that point. A second interesting observation, is that Theorem 1 can be evoked to show that the posterior is Gaussian whenever all high-order derivatives of $\mu_1(\cdot)$ vanish (see proof in App. F).

**Corollary 1.** *Assume that* $\mu_1^{(k)}(y^*) = 0$ *for all* $k > 1$. *Then the posterior* $p_{x|y}(\cdot|y^*)$ *is Gaussian.*

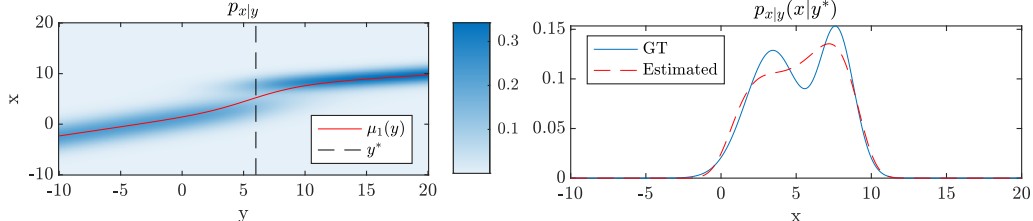

Figure 1: **Recovering posteriors in univariate denoising**. The left pane shows the posterior distribution $p_{x|y}(\cdot|\cdot)$ and the posterior mean function $\mu_1(\cdot)$ for the scalar Gaussian denoising task (1). On the right we plot the posterior distribution of x given that $y = y^*$, along with an estimate of that distribution, which we obtain by analyzing the denoiser function $\mu_1(\cdot)$ at the vicinity of $y^*$. Specifically, this estimate corresponds to the maximum entropy distribution that matches the first four moments, which are obtained from Theorem 1 by numerically approximating $\mu_1'(y^*), \mu_1''(y^*), \mu_1'''(y^*)$.

## 3.2 THE MULTIVARIATE CASE

We now move on to treat the multivariate denoising problem. Here $\mathbf{x}$ is a random vector taking values in $\mathbb{R}^d$, the noise $\mathbf{n} \sim \mathcal{N}(0, \sigma^2 \boldsymbol{I}_d)$ is a white multivariate Gaussian vector that is statistically independent of $\mathbf{x}$, and the noisy observation is

$$\mathbf{y} = \mathbf{x} + \mathbf{n}. \tag{5}$$

As in the scalar setting, given a noisy measurement $\mathbf{y} = \boldsymbol{y}$, we are interested in the posterior distribution $p_{\mathbf{x}|\mathbf{y}}(\cdot|\boldsymbol{y})$. The MSE-optimal denoiser is, again, the first-order moment of this distribution,

$$\boldsymbol{\mu}_1(\boldsymbol{y}) = \mathbb{E}[\mathbf{x} \,|\, \mathbf{y} = \boldsymbol{y}], \tag{6}$$

which is a $d$ dimensional vector. The second-order central moment is the posterior covariance

$$\boldsymbol{\mu}_2(\boldsymbol{y}) = \mathrm{Cov}(\mathbf{x} \,|\, \mathbf{y} = \boldsymbol{y}), \tag{7}$$

which is a $d \times d$ matrix whose $(i_1, i_2)$ entry is given by

$$[\boldsymbol{\mu}_2(\boldsymbol{y})]_{i_1, i_2} = \mathbb{E}\left[(\mathbf{x}_{i_1} - [\mu_1(\mathbf{y})]_{i_1})(\mathbf{x}_{i_2} - [\mu_1(\mathbf{y})]_{i_2}) \,|\, \mathbf{y} = \boldsymbol{y}\right]. \tag{8}$$

For any $k \geq 3$, the posterior $k$th-order central moment is a $d \times \cdots \times d$ array with $k$ indices (a $k$th order tensor), whose component at multi-index $(i_1, \ldots, i_k)$ is given by

$$[\boldsymbol{\mu}_k(\boldsymbol{y})]_{i_1, \ldots, i_k} = \mathbb{E}\left[(\mathbf{x}_{i_1} - [\mu_1(\mathbf{y})]_{i_1}) \cdots (\mathbf{x}_{i_k} - [\mu_1(\mathbf{y})]_{i_k}) \,|\, \mathbf{y} = \boldsymbol{y}\right]. \tag{9}$$

As we now show, similarly to the scalar case, having access to the MSE-optimal denoiser and its derivatives, allows to recursively compute all higher order posterior moments (see proof in App. B).

**Theorem 2** (Posterior moments in multivariate denoising). *Consider the multivariate denoising setting of (5) with dimension $d \geq 2$. For any $k \geq 1$ and any $k+1$ indices $i_1, \ldots, i_{k+1} \in \{1, \ldots, d\}$, the high-order posterior central moments of $\mathbf{x}$ given $\mathbf{y}$ satisfy the recursion*

$$[\boldsymbol{\mu}_2(\boldsymbol{y})]_{i_1, i_2} = \sigma^2 \frac{\partial [\boldsymbol{\mu}_1(\boldsymbol{y})]_{i_1}}{\partial \boldsymbol{y}_{i_2}},$$

$$[\boldsymbol{\mu}_3(\boldsymbol{y})]_{i_1, i_2, i_3} = \sigma^2 \frac{\partial [\boldsymbol{\mu}_2(\boldsymbol{y})]_{i_1, i_2}}{\partial \boldsymbol{y}_{i_3}},$$

$$[\boldsymbol{\mu}_{k+1}(\boldsymbol{y})]_{i_1, \ldots, i_{k+1}} = \sigma^2 \frac{\partial [\boldsymbol{\mu}_k(\boldsymbol{y})]_{i_1, \ldots, i_k}}{\partial \boldsymbol{y}_{i_{k+1}}} + \sum_{j=1}^{k} [\boldsymbol{\mu}_{k-1}(\boldsymbol{y})]_{\boldsymbol{\ell}_j} [\boldsymbol{\mu}_2(\boldsymbol{y})]_{i_j, i_{k+1}}, \quad k \geq 3, \tag{10}$$

*where $\boldsymbol{\ell}_j \triangleq (i_1, \ldots, i_{j-1}, i_{j+1} \ldots, i_k)$. Thus, $\boldsymbol{\mu}_{k+1}(\boldsymbol{y})$ is uniquely determined by $\boldsymbol{\mu}_1(\boldsymbol{y})$ and by the derivatives up to order $k$ of its elements with respect to the elements of the vector $\boldsymbol{y}$.*

Note that the first line in (10) can be compactly written as

$$\boldsymbol{\mu}_2(\boldsymbol{y}) = \sigma^2 \frac{\partial \boldsymbol{\mu}_1(\boldsymbol{y})}{\partial \boldsymbol{y}}, \tag{11}$$

---

**Algorithm 1** Efficient computation of posterior principal components

**Input:** $N$ (number of PCs), $K$ (number of iterations), $\boldsymbol{\mu}_1(\cdot)$ (MSE-optimal denoiser), $\boldsymbol{y}$ (noisy input), $\sigma^2$ (noise variance), $c \ll 1$ (linear approx. constant)

1: Initialize $\{\boldsymbol{v}_0^{(i)}\}_{i=1}^N \leftarrow \mathcal{N}(0, \sigma^2 \boldsymbol{I})$
2: **for** $k \leftarrow 1$ to $K$ **do**
3:     **for** $i \leftarrow 1$ to $N$ **do**
4:         $\boldsymbol{v}_k^{(i)} \leftarrow \frac{1}{c}\left(\boldsymbol{\mu}_1(\boldsymbol{y} + c\boldsymbol{v}_{k-1}^{(i)}) - \boldsymbol{\mu}_1(\boldsymbol{y})\right)$
5:     $\mathbf{Q}, \mathbf{R} \leftarrow \text{QR\_DECOMPOSITION}([\boldsymbol{v}_k^{(1)} \cdots \boldsymbol{v}_k^{(N)}])$
6:     $[\boldsymbol{v}_k^{(1)} \cdots \boldsymbol{v}_k^{(N)}] \leftarrow \mathbf{Q}$
7: $\boldsymbol{v}^{(i)} \leftarrow \boldsymbol{v}_K^{(i)}$
8: $\lambda^{(i)} \leftarrow \frac{\sigma^2}{c}\|\boldsymbol{\mu}_1(\boldsymbol{y} + c\boldsymbol{v}_{K-1}^{(i)}) - \boldsymbol{\mu}_1(\boldsymbol{y})\|$

---

where $\frac{\partial \boldsymbol{\mu}_1(\boldsymbol{y})}{\partial \boldsymbol{y}}$ denotes the Jacobian of $\boldsymbol{\mu}_1$ at $\boldsymbol{y}$. This suggests that, in principle, the posterior covariance of an MSE-optimal denoiser could be extracted by computing the Jacobian of the model using *e.g.*, automatic differentiation. However, in settings involving high-resolution images, even storing this Jacobian is impractical. In Sec. 4.1, we show how the top eigenvectors of $\boldsymbol{\mu}_2(\boldsymbol{y})$ (*i.e.*, the posterior principal components) can be computed without having to ever store $\boldsymbol{\mu}_2(\boldsymbol{y})$ in memory.

Moments of order greater than two pose an even bigger challenge, as they correspond to higher-order tensors. In fact, even if they could somehow be computed, it is not clear how they would be visualized in order to communicate the uncertainty of the prediction to a user. A practical solution could be to visualize the posterior distribution of the projection of $\mathbf{x}$ onto some meaningful one-dimensional space. For example, one might be interested in the posterior distribution of $\mathbf{x}$ projected onto one of the principal components of the posterior covariance. The question, however, is how to obtain the posterior moments of the projection of $\mathbf{x}$ onto a deterministic $d$-dimensional vector $\boldsymbol{v}$.

Let us denote the first posterior moment of $\boldsymbol{v}^\top \mathbf{x}$ (*i.e.*, its posterior mean) by $\mu_1^{\boldsymbol{v}}(\boldsymbol{y})$. This moment is given by the projection of the denoiser's output onto $\boldsymbol{v}$,

$$\mu_1^{\boldsymbol{v}}(\boldsymbol{y}) = \mathbb{E}\left[\boldsymbol{v}^\top \mathbf{x}\big|\mathbf{y} = \boldsymbol{y}\right] = \boldsymbol{v}^\top \mathbb{E}\left[\mathbf{x}|\mathbf{y} = \boldsymbol{y}\right] = \boldsymbol{v}^\top \boldsymbol{\mu}_1(\boldsymbol{y}). \tag{12}$$

Similarly, let us denote the $k$th order posterior central moment of $\boldsymbol{v}^\top \mathbf{x}$ by

$$\mu_k^{\boldsymbol{v}}(\boldsymbol{y}) = \mathbb{E}\left[\left(\boldsymbol{v}^\top \mathbf{x} - \boldsymbol{v}^\top \boldsymbol{\mu}_1(\boldsymbol{y})\right)^k\Big|\mathbf{y} = \boldsymbol{y}\right], \quad k \geq 2. \tag{13}$$

As we show next, the scalar-valued functions $\{\mu_k^{\boldsymbol{v}}(\boldsymbol{y})\}_{k=1}^\infty$ satisfy a recursion similar to (4) (see proof in App. C). In Sec. 5, we use this result for uncertainty visualization.

**Theorem 3** (Directional posterior moments in multivariate denoising)**.** *Let $\boldsymbol{v}$ be a deterministic $d$-dimensional vector. Then the posterior central moments of $\boldsymbol{v}^\top \mathbf{x}$ are given by the recursion*

$$\mu_2^{\boldsymbol{v}}(\boldsymbol{y}) = \sigma^2 D_{\boldsymbol{v}}\mu_1^{\boldsymbol{v}}(\boldsymbol{y}),$$
$$\mu_3^{\boldsymbol{v}}(\boldsymbol{y}) = \sigma^2 D_{\boldsymbol{v}}\mu_2^{\boldsymbol{v}}(\boldsymbol{y}),$$
$$\mu_{k+1}^{\boldsymbol{v}}(\boldsymbol{y}) = \sigma^2 D_{\boldsymbol{v}}\mu_k^{\boldsymbol{v}}(\boldsymbol{y}) + k\mu_{k-1}^{\boldsymbol{v}}(\boldsymbol{y})\mu_2^{\boldsymbol{v}}(\boldsymbol{y}), \quad k \geq 3. \tag{14}$$

*Here $D_{\boldsymbol{v}} f(\boldsymbol{y})$ denotes the directional derivative of a function $f : \mathbb{R}^d \to \mathbb{R}$ in direction $\boldsymbol{v}$ at $\boldsymbol{y}$.*

## 4 APPLICATION TO UNCERTAINTY VISUALIZATION

We now discuss the applicability of our results in the context of uncertainty visualization. We start with efficient computation of posterior principal components (PCs), and then illustrate the approximation of marginal densities along those directions.

### 4.1 EFFICIENT COMPUTATION OF POSTERIOR PRINCIPAL COMPONENTS

The top eigenvectors of the posterior covariance, $\boldsymbol{\mu}_2(\boldsymbol{y})$, capture the main modes of variation around the MSE-optimal prediction. Thus, as we illustrate below, they reveal meaningful information regarding the uncertainty of the restoration. Had we had access to the matrix $\boldsymbol{\mu}_2(\boldsymbol{y})$, computing these

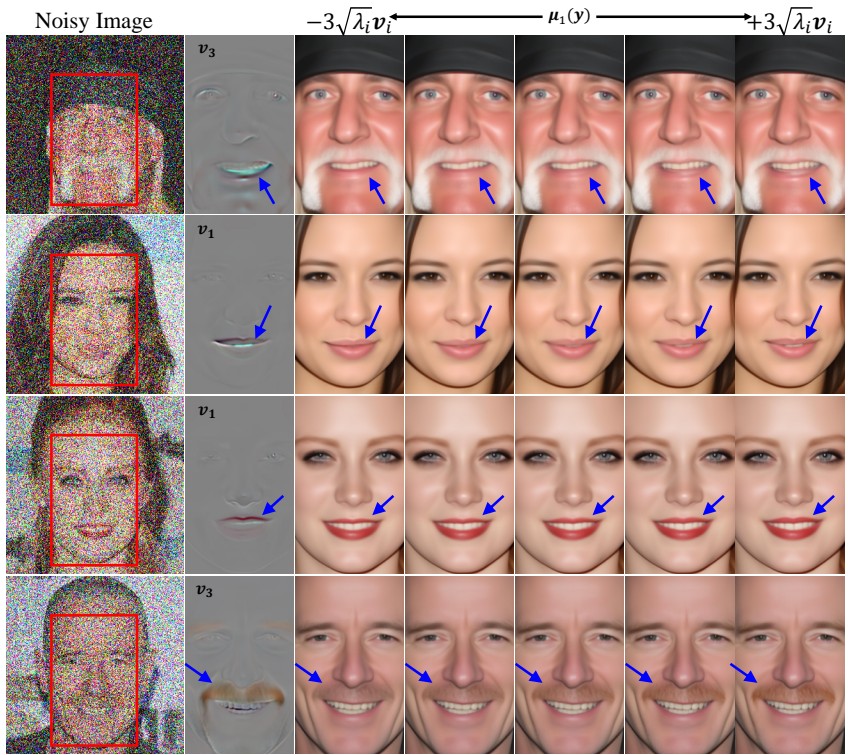

Figure 2: **Computing posterior principal components for a pre-trained face denoising model**. For each noisy image $\boldsymbol{y}$, we depict one of the posterior PCs obtained with Alg. 1. To the right of that PC, we show the denoiser's output, $\boldsymbol{\mu}_1(\boldsymbol{y})$, and its perturbation along that PC. As can be seen, this visualization captures the denoiser's uncertainty along semantically meaningful directions, such as the color of the moustache, the thickness of the lips, and the extent to which the mouth is open.

top eigenvectors could be done using the subspace iteration method (Arbenz, 2016; Saad, 2011). This technique maintains a set of $N$ vectors, which are repeatedly multiplied by $\boldsymbol{\mu}_2(\boldsymbol{y})$ and orthonormalized using the QR decomposition. Unfortunately, storing the full covariance matrix is commonly impractical. To circumvent the need for doing so, we recall from (11) that $\boldsymbol{\mu}_2(\boldsymbol{y})$ corresponds to the Jacobian of the denoiser $\boldsymbol{\mu}_1(\boldsymbol{y})$. Thus, every iteration of the subspace method corresponds to a Jacobian-vector product. For neural denoisers, such products can be calculated using automatic differentiation (Dockhorn et al., 2022). However, this requires computing a backward pass through the model in each iteration, which can become computationally demanding for large images[1]. Instead, we propose to use the linear approximation

$$\frac{\partial \boldsymbol{\mu}_1(\boldsymbol{y})}{\partial \boldsymbol{y}} \boldsymbol{v} \approx \frac{\boldsymbol{\mu}_1(\boldsymbol{y} + c\boldsymbol{v}) - \boldsymbol{\mu}_1(\boldsymbol{y})}{c}, \tag{15}$$

which holds for any $\boldsymbol{v} \in \mathbb{R}^d$ when $c \in \mathbb{R}$ is sufficiently small. This allows applying the subspace iteration using only forward passes through the denoiser, as summarized in Alg. 1. As we show in App. H, this approximation has a negligible effect on the calculated eigenvectors, but leads *e.g.*, to a $6\times$ reduction in memory footprint for a $80 \times 92$ patch with the SwinIR denoiser (Liang et al., 2021). We note that to compute the PCs for a user-chosen region of interest, all that is required is to mask out all entries of $\boldsymbol{v}$ outside that region in each iteration.

Figure 2 illustrates this technique in the context of denoising of face images contaminated by white Gaussian noise with standard deviation $\sigma = 122$. We use the denoiser from (Baranchuk et al., 2022), which was trained as part of a DDPM model (Ho et al., 2020) on the FFHQ dataset (Karras et al., 2019). Note that here we use it as a plain denoiser (as used within a single timestep of the

---

[1]Note that backward passes for whole images are also often avoided during training of neural denoisers. Indeed, typical training procedures use limited-sized crops.

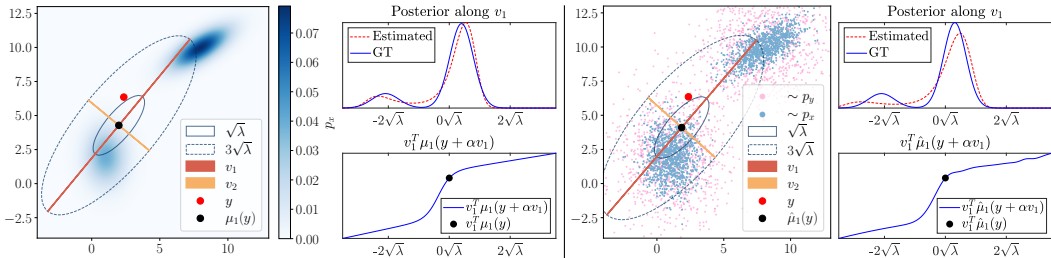

Figure 3: **Computing marginals along principal components**. On the left, we show the prior $p_{\mathbf{x}}$ as a heatmap, a noisy sample $\boldsymbol{y}$ (red), the corresponding MSE-optimal estimate $\boldsymbol{\mu}_1(\boldsymbol{y})$ (black), and the two principal axes, computed using Alg. 1. Here, we used the closed form for $\boldsymbol{\mu}_1(\boldsymbol{y})$. The second pane shows the marginal posterior distribution along the first principal component, computed both using our proposed procedure (dashed red), and by using the closed-form solution (solid blue). On the right we show the same experiment, but with a simple neural network trained on data samples.

DDPM). We showcase examples from the CelebAMask-HQ dataset (Lee et al., 2020). As can be seen, different posterior principal components typically capture uncertainty in different localized regions of the image. Note that this approach can be applied to any region-of-interest within the image, chosen by the user at test time. This is in contrast to a model that is trained to predict a low-rank approximation of the covariance, as in (Meng et al., 2021). Such a model is inherently limited to the specific input size on which it was trained, and cannot be manipulated at test time to produce eigenvectors corresponding to some user-chosen region (cropping a patch from an eigenvector is not equivalent to computing the eigenvector of the corresponding patch in the image). In App. K we report quantitative comparisons to the naive baseline of estimating the PCs using a posterior sampler, and quantitatively evaluate the accuracy of the eigenvalues predicted by our method.

## 4.2 ESTIMATION OF MARGINAL DISTRIBUTIONS ALONG CHOSEN DIRECTIONS

A more fine-grained characterization of the posterior can be achieved by using higher-order moments along the principal directions. These can be calculated using Theorem 3, through (high-order) numerical differentiation of the one-dimensional function $f(\alpha) = \boldsymbol{v}^\top \boldsymbol{\mu}_1(\boldsymbol{y} + \alpha \boldsymbol{v})$ at $\alpha = 0$. Once we obtain all moments up to some order, we compute the probability distribution with maximum entropy that fits those moments. In practice, we compute derivatives up to third order, which allows us to obtain all moments up to order four.

Figure 3 illustrates this approach on a two-dimensional Gaussian mixture example with a noise level of $\sigma = 2$. On the left, we show a heatmap corresponding to $p_{\mathbf{x}}(\cdot)$, as well as a noisy input $\boldsymbol{y}$ (red point) and its corresponding MSE-optimal estimate (black point). The two axes of the ellipse are the posterior principal components computed using Alg. 1 using numerical differentiation of the closed-form expression of the denoiser (see App. E). The bottom plot on the second pane shows the function $f(\alpha)$ corresponding to the largest eigenvector. We numerically computed its derivatives up to order three at $\alpha = 0$ (black point), from which we estimated the moments up to order four according to Theorem 3. The top plot on that pane shows the ground-truth posterior distribution of $\boldsymbol{v}_1^\top \mathbf{x}$, along with the maximum entropy distribution computed from the moments. The right half of the figure shows the same experiment only with a neural network that was trained on pairs of noisy (pink) samples and their clean (blue) counterparts. This denoiser comprises 5 layers with $(100, 200, 200, 100)$ hidden features and SiLU (Hendrycks & Gimpel, 2016) activation units. We trained the network using Adam (Kingma & Ba, 2015) for 300 epochs, with a learning rate of 0.005.

Figure 4 illustrates the approach on a handwritten digit from the MNIST (LeCun, 1998) dataset. Here, we train and use a simple CNN with 10 layers of 64 channels, separated by ReLU activation layers followed by batch normalization layers. As can be seen, fitting the maximum entropy distribution reveals more than just the main modes of variation, as it also reveals the likelihood of each reconstruction along that direction. It is instructive to note that although the two extreme reconstructions, $\boldsymbol{\mu}_1(\boldsymbol{y}) \pm \sqrt{\lambda_3} \boldsymbol{v}_3$, look realistic, they are not probable given the noisy observation. This is the

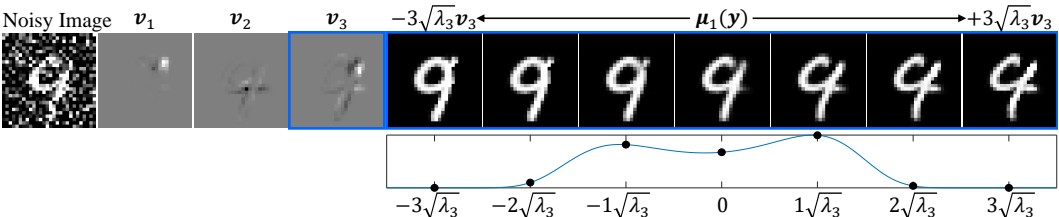

Figure 4: **Uncertainty quantification for denoising a handwritten digit**. The first three PCs corresponding to the noisy image are shown on the left. On the right, images along the third PC, marked in blue, are shown, together with the marginal posterior distribution we estimated for that direction. The two modes of the possible restoration, corresponding to the digits 4 and 9, can clearly be seen as peaks in the marginal posterior distribution, whereas the MSE-optimal restoration in the middle is obviously less likely.

reason their corresponding estimated posterior density is nearly zero. In App. K, we quantitatively validate the advantage of using higher-order moments for estimating the marginal distribution.

Our theoretical analysis applies to non-blind denoising, in which $\sigma$ is known. However, we empirically show in Sec. 5 and Fig. 5 that using an estimated $\sigma$ is also sufficient for obtaining qualitatively plausible results. This can either be obtained from a noise estimation method (Chen et al., 2015) or even from the naive estimate $\hat{\sigma}^2 = \frac{1}{d}\|\boldsymbol{\mu}_1(\boldsymbol{y}) - \boldsymbol{y}\|^2$, where $\boldsymbol{\mu}_1(\boldsymbol{y})$ is the output of a blind denoiser. Here we use the latter. We further discuss the impact of using an estimated $\sigma$ in App. I.

## 5 EXPERIMENTS

We conduct experiments with our proposed approach for uncertainty visualization and marginal posterior distribution estimation on additional real data in multiple domains using different models.

We showcase our method on the MNIST dataset, natural images, human faces, and on images from the microscopy domain. For natural images, we use SwinIR (Liang et al., 2021) that was pre-trained on 800 DIV2K (Agustsson & Timofte, 2017) images, 2650 Flickr2k (Lim et al., 2017) images, 400 BSD500 (Arbelaez et al., 2010) images and 4,744 WED (Ma et al., 2016) images, with patch sizes $128 \times 128$ and window size $8 \times 8$. We experiment with two SwinIR models, trained separately for noise levels $\sigma = \{25, 50\}$, and showcase examples on test images from the CBSD68 (Martin et al., 2001) and Kodak (Franzen, 1999) datasets. For the medical and microscopy domain we use Noise2Void (Krull et al., 2019), trained and tested for blind-denoising on the FMD dataset (Zhang et al., 2019) in the unsupervised manner described by Krull et al. (2020). The FMD dataset was collected using real microscopy imaging, and as such its noise is most probably not precisely white nor Gaussian, and the noise level is unknown in essence (the ground truth images are considered as the average of 50 burst images). Accordingly, N2V is a blind-denoiser, and we have no access to the "real" $\sigma$, therefore, for this dataset we used an estimated $\sigma$ in our method, as described in Sec. 4.2.

Examples for the different domains can be seen in Figs. 2, 4, and 5. As can be seen, in all cases, our approach captures interesting uncertainty directions. For natural images, those include cracks, wrinkles, eye colors, stripe shapes, etc. In the biological domain, visualizations reveal uncertainty in the size and morphology of cells, as well as in the (in)existence of septums. Those constitute important geometric features in cellular analysis. More examples can be found in App. L.

One limitation of the proposed method is that it relies on high-order numerical differentiation. As this approximation can be unstable with low-precision computation, we use double precision during the forward pass of the networks. Another method that can be used to mitigate this is to fit a low degree polynomial to $f(\alpha) = \boldsymbol{v}^\top \boldsymbol{\mu}_1(\boldsymbol{y} + \alpha \boldsymbol{v})$ around the point of derivation, $\alpha = 0$, and then use the smooth polynomial fit for the high-order derivatives calculation. Empirically we found the polynomial fitting to also be sensitive, highly-dependant on the choice of the polynomial degree and the fitted range. This caused bad fits even for the simple two-component GMM example, whereas the numerical derivatives approximations worked better.

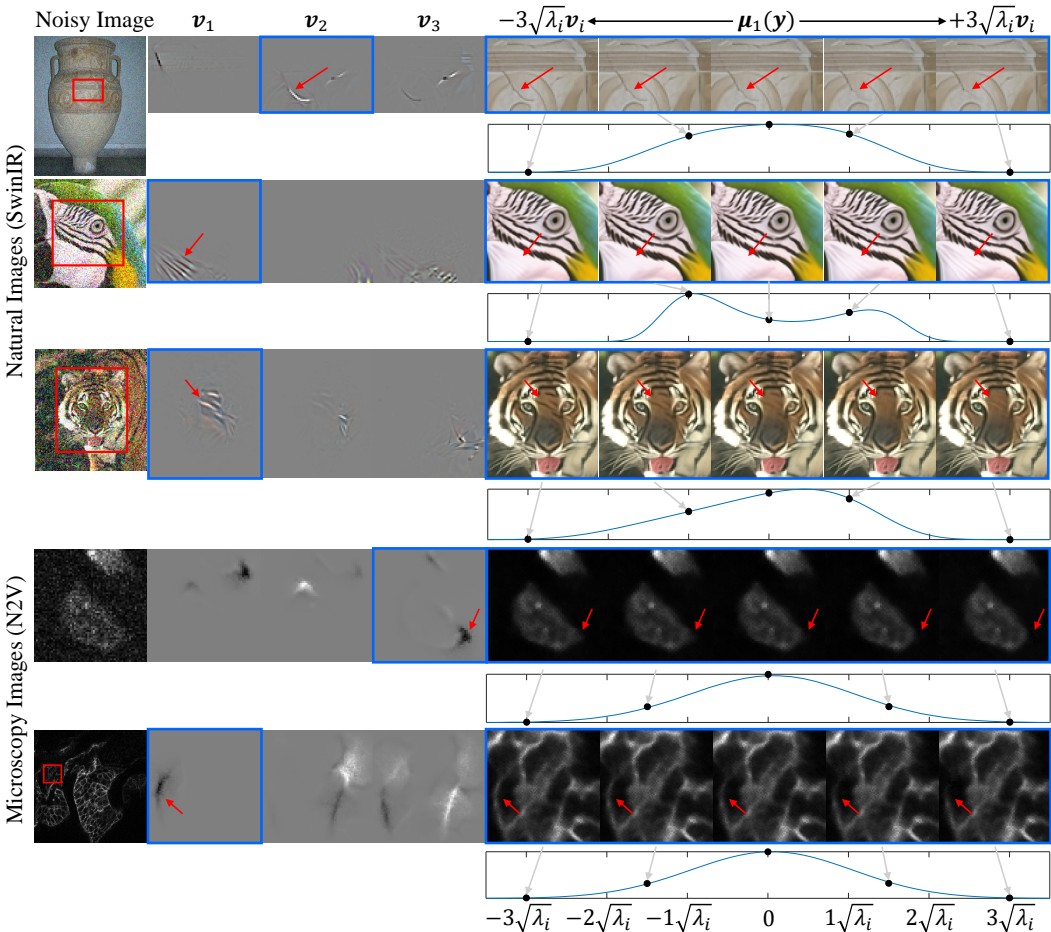

Figure 5: **Uncertainty quantification for natural image denoising using SwinIR (top) and microscopy image denoising using N2V (bottom)**. In each row, the first three PCs corresponding to the noisy image are shown on the left, and one is marked in blue. On the right, images along the marked PC are shown above the marginal posterior distribution estimated for this direction. The PCs show the uncertainty along meaningful directions, such as the existence of cracks on an old vase and changes in the tiger's stripes, as well as the sizes of cells and the existence of septum, which constitute important geometric features in cellular analysis.

## 6 Conclusion

Denoisers constitute fundamental ingredients in a variety of problems. In this paper we derived a relation in the denoising problem between higher-order derivatives of the posterior mean to higher-order posterior central moments. These results were then used in the application of uncertainty visualisation of pre-trained denoisers. Specifically, we proposed a method for efficiently computing the principal components of the posterior distribution, in any chosen region of an image. Additionally, we presented a scheme to use higher-order moments to estimate the full marginal distribution along any one-dimensional direction. Finally, we demonstrated our method on multiple denoisers across different domains. Our method allows examining semantic directions of uncertainty by using only pre-trained denoisers, in a fast and memory-efficient way. While the theoretical basis of our method applies only to additive white Gaussian noise, we show empirically that our method provides qualitatively satisfactory results also in blind denoising on real-world microscopy data.

## REPRODUCIBILITY STATEMENT

As part of the ongoing effort to make the field of deep learning more reproducible and open, we publish our code at `https://hilamanor.github.io/GaussianDenoisingPosterior/`. The repository includes scripts to regenerate all figures. Researchers that want to re-implement the code from scratch can use Alg. 1 and our published code as guidelines. In addition, we provide full and detailed proofs for all claims in the paper in Appendices A, B, C, E, and F of the supplementary material. Finally, we provide in Appendix D a translation from our notation to the notation of Meng et al. (2021) to allow future researchers to use both methods conveniently.

## ETHICS STATEMENT

In many scientific and medical domains, signals are contaminated by noise, and deep learning based denoising models have emerged as popular tools for restoring such low-fidelity data. However, denoising problems are inherently ill-posed. Therefore, a system that presents users with only a single restored signal, may mislead the data-analyst, researcher, or physician into making flawed decisions. To avoid such situations, it is of utmost importance for systems to also report and conveniently visualize the uncertainties in their predictions. Such systems would be much more trustworthy and interpretable, and will thus support making credible deductions and decisions. The method we presented in this paper, can help visualize the uncertainty in a denoiser's prediction, by allowing users to explore the dominant modes of possible variations around that prediction, accompanied by their likelihood (given the noisy measurements). Such interactive denoising systems, would allow users to take into consideration other, and sometimes even more likely possibilities than *e.g.*, the minimum MSE reconstruction that is often reported as a single solution.

### ACKNOWLEDGEMENTS

The Miriam and Aaron Gutwirth Memorial Fellowship supported the research of HM. The research of TM was supported by the Israel Science Foundation (grant no. 2318/22), by the Ollendorff Miverva Center, ECE faculty, Technion, and by a gift from Elbit. The authors are grateful to Elias Nehme, Rotem Mulayoff, and Matan Kleiner for their insightful discussions and input throughout this work, and Noa Cohen for her invaluable help with the algorithm.

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

# SUPPLEMENTARY MATERIAL

## A  PROOF OF THEOREM 1

We start with the case $k \geq 2$ (bottom two lines in (4)). In this case, the conditional moment $\mu_k(y)$ can be expressed using Bayes' formula as

$$
\begin{aligned}
\mu_k(y) &= \mathbb{E}\left[(\mathrm{x} - \mu_1(\mathrm{y}))^k \,\middle|\, \mathrm{y} = y\right] \\
&= \int (x - \mu_1(y))^k p_{\mathrm{x}|\mathrm{y}}(x|y) dx \\
&= \frac{\int (x - \mu_1(y))^k p_{\mathrm{y}|\mathrm{x}}(y|x) p_{\mathrm{x}}(x) dx}{p_{\mathrm{y}}(y)} \\
&= \frac{(2\pi\sigma^2)^{-\frac{1}{2}} \int (x - \mu_1(y))^k \exp\{-\frac{1}{2\sigma^2}(y - x)^2\} p_{\mathrm{x}}(x) dx}{p_{\mathrm{y}}(y)}.
\end{aligned}
\tag{S1}
$$

Denoting the numerator by $q(y) \triangleq (2\pi\sigma^2)^{-\frac{1}{2}} \int (x - \mu_1(y))^k \exp\{-\frac{1}{2\sigma^2}(y - x)^2\} p_{\mathrm{x}}(x) dx$, we can write the derivative of $\mu_k(y)$ as

$$
\begin{aligned}
\mu_k'(y) &= \frac{q'(y)p_{\mathrm{y}}(y) - q(y)p_{\mathrm{y}}'(y)}{p_{\mathrm{y}}^2(y)} \\
&= \frac{q'(y)}{p_{\mathrm{y}}(y)} - \frac{q(y)}{p_{\mathrm{y}}(y)} \frac{p_{\mathrm{y}}'(y)}{p_{\mathrm{y}}(y)} \\
&= \frac{q'(y)}{p_{\mathrm{y}}(y)} - \mu_k(y) \frac{p_{\mathrm{y}}'(y)}{p_{\mathrm{y}}(y)} \\
&= \frac{q'(y)}{p_{\mathrm{y}}(y)} - \mu_k(y) \frac{d \log p_{\mathrm{y}}(y)}{dy} \\
&= \frac{q'(y)}{p_{\mathrm{y}}(y)} - \frac{1}{\sigma^2} \mu_k(y)(\mu_1(y) - y),
\end{aligned}
\tag{S2}
$$

where we used the fact that $\frac{d \log p_{\mathrm{y}}(y)}{dy} = \frac{1}{\sigma^2}(\mu_1(y) - y)$ (see $e.g.$, (Efron, 2011; Miyasawa et al., 1961; Stein, 1981)). The first term in this expression is given by

$$
\begin{aligned}
\frac{q'(y)}{p_{\mathrm{y}}(y)} &= \frac{(2\pi\sigma^2)^{-\frac{1}{2}} \int \frac{d}{dy}\left[(x - \mu_1(y))^k \exp\{-\frac{1}{2\sigma^2}(y - x)^2\}\right] p_{\mathrm{x}}(x) dx}{p_{\mathrm{y}}(y)} \\
&= \frac{(2\pi\sigma^2)^{-\frac{1}{2}} \int \left(-k(x - \mu_1(y))^{k-1}\mu_1'(y) - (x - \mu_1(y))^k \frac{1}{\sigma^2}(y - x)\right) \exp\{-\frac{1}{2\sigma^2}(y - x)^2\} p_{\mathrm{x}}(x) dx}{p_{\mathrm{y}}(y)} \\
&= \frac{\int \left(-k(x - \mu_1(y))^{k-1}\mu_1'(y) - (x - \mu_1(y))^k \frac{1}{\sigma^2}(y - x)\right) p_{\mathrm{y}|\mathrm{x}}(y|x) p_{\mathrm{x}}(x) dx}{p_{\mathrm{y}}(y)} \\
&= \int \left(-k(x - \mu_1(y))^{k-1}\mu_1'(y) - (x - \mu_1(y))^k \frac{1}{\sigma^2}(y - x)\right) p_{\mathrm{x}|\mathrm{y}}(x|y) dx \\
&= \mathbb{E}\left[-k(\mathrm{x} - \mu_1(\mathrm{y}))^{k-1}\mu_1'(\mathrm{y}) - (\mathrm{x} - \mu_1(\mathrm{y}))^k \frac{1}{\sigma^2}(\mathrm{y} - \mathrm{x}) \,\middle|\, \mathrm{y} = y\right].
\end{aligned}
\tag{S3}
$$

To allow unified treatment of the cases $k = 2$ and $k > 2$, let us denote

$$
\psi_k(y) \triangleq \mathbb{E}\left[(\mathrm{x} - \mu_1(\mathrm{y}))^k \,\middle|\, \mathrm{y} = y\right] = \begin{cases} 0 & k = 1, \\ \mu_k(y) & k \geq 2. \end{cases}
\tag{S4}
$$

We therefore have

$$\frac{q'(y)}{p_y(y)} = -k\psi_{k-1}(y)\mu_1'(y) - \frac{1}{\sigma^2}\psi_k(y)y + \frac{1}{\sigma^2}\mathbb{E}\left[(\mathrm{x} - \mu_1(\mathrm{y}))^k\mathrm{x} \,\middle|\, \mathrm{y} = y\right]$$

$$= -k\psi_{k-1}(y)\mu_1'(y) - \frac{1}{\sigma^2}\psi_k(y)y + \frac{1}{\sigma^2}\mathbb{E}[(\mathrm{x} - \mu_1(\mathrm{y}))^k(\mathrm{x} - \mu_1(\mathrm{y}) + \mu_1(\mathrm{y})) \,|\, \mathrm{y} = y]$$

$$= -k\psi_{k-1}(y)\mu_1'(y) - \frac{1}{\sigma^2}\psi_k(y)y + \frac{1}{\sigma^2}\left(\psi_{k+1}(y) + \psi_k(y)\mu_1(y)\right)$$

$$= -k\psi_{k-1}(y)\mu_1'(y) + \frac{1}{\sigma^2}\psi_{k+1}(y) + \frac{1}{\sigma^2}\psi_k(y)\left(\mu_1(y) - y\right). \tag{S5}$$

Substituting this back into (S2), we obtain that

$$\mu_k'(y) = -k\psi_{k-1}(y)\mu_1'(y) + \frac{1}{\sigma^2}\psi_{k+1}(y) + \frac{1}{\sigma^2}\psi_k(y)\left(\mu_1(y) - y\right) - \frac{1}{\sigma^2}\mu_k(y)\left(\mu_1(y) - y\right)$$

$$= -k\psi_{k-1}(y)\mu_1'(y) + \frac{1}{\sigma^2}\psi_{k+1}(y), \tag{S6}$$

where we used the fact that $\psi_k(y) = \mu_k(y)$ for all $k \geq 2$. Now, for $k = 2$ this equation reads

$$\mu_2'(y) = \frac{1}{\sigma^2}\mu_3(y), \tag{S7}$$

and for $k \geq 3$, it reads

$$\mu_k'(y) = -k\mu_{k-1}(y)\mu_1'(y) + \frac{1}{\sigma^2}\mu_{k+1}(y). \tag{S8}$$

We thus have that

$$\mu_3(y) = \sigma^2\mu_2'(y),$$
$$\mu_{k+1}(y) = \sigma^2\mu_k'(y) + k\sigma^2\mu_{k-1}(y)\mu_1'(y), \quad k \geq 3. \tag{S9}$$

Note that an equivalent expression for the last line is obtained by replacing $\sigma^2\mu_1'(y)$ with $\mu_2(y)$, as we prove below. This completes the proof for $k \geq 2$.

The case $k = 1$ can be treated similarly. Here,

$$\mu_1(y) = \mathbb{E}[\mathrm{x} \,|\, \mathrm{y} = y]$$

$$= \frac{(2\pi\sigma^2)^{-\frac{1}{2}}\int x\exp\{-\frac{1}{2\sigma^2}(y-x)^2\}p_x(x)dx}{p_y(y)}, \tag{S10}$$

so that we define $q(y) \triangleq (2\pi\sigma^2)^{-\frac{1}{2}}\int x\exp\{-\frac{1}{2\sigma^2}(y-x)^2\}p_x(x)dx$. We thus have

$$\frac{q'(y)}{p_y(y)} = \frac{(2\pi\sigma^2)^{-\frac{1}{2}}\int \frac{d}{dy}\left[x\exp\{-\frac{1}{2\sigma^2}(y-x)^2\}\right]p_x(x)dx}{p_y(y)}$$

$$= \frac{(2\pi\sigma^2)^{-\frac{1}{2}}\int \frac{1}{\sigma^2}(x-y)\exp\{-\frac{1}{2\sigma^2}(y-x)^2\}p_x(x)dx}{p_y(y)}$$

$$= \frac{1}{\sigma^2}\mathbb{E}[\mathrm{x}(\mathrm{x} - \mathrm{y}) \,|\, \mathrm{y} = y]$$

$$= \frac{1}{\sigma^2}\left(\mathbb{E}[\mathrm{x}^2 \,|\, \mathrm{y} = y] - \mu_1(y)y\right). \tag{S11}$$

Therefore,

$$\mu_1'(y) = \frac{q'(y)}{p_y(y)} - \frac{1}{\sigma^2}\mu_1(y)(\mu_1(y) - y)$$

$$= \frac{1}{\sigma^2}\left(\mathbb{E}[\mathrm{x}^2 \,|\, \mathrm{y} = y] - \mu_1(y)y\right) - \frac{1}{\sigma^2}\mu_1(y)(\mu_1(y) - y)$$

$$= \frac{1}{\sigma^2}\left(\mathbb{E}[\mathrm{x}^2 \,|\, \mathrm{y} = y] - \mu_1^2(y)\right)$$

$$= \frac{1}{\sigma^2}\left(\mathbb{E}[\mathrm{x}^2 \,|\, \mathrm{y} = y] - \mathbb{E}[\mathrm{x} \,|\, \mathrm{y} = y]^2\right)$$

$$= \frac{1}{\sigma^2}\mu_2(y), \tag{S12}$$

which demonstrates that

$$\mu_2(y) = \sigma^2 \mu_1'(y). \tag{S13}$$

This completes the proof for $k = 1$.

## B  PROOF OF THEOREM 2

We begin with the case $k = 1$ (first line in (10)), by directly deriving the matrix form (11). Using Bayes' formula, the posterior mean $\boldsymbol{\mu}_1(\boldsymbol{y})$ can be expressed as

$$
\begin{aligned}
\boldsymbol{\mu}_1(\boldsymbol{y}) &= \mathbb{E}[\mathbf{x}|\mathbf{y} = \boldsymbol{y}] \\
&= \int_{\mathbb{R}^d} \boldsymbol{x} p_{\mathbf{x}|\mathbf{y}}(\boldsymbol{x}|\boldsymbol{y}) d\boldsymbol{x} \\
&= \frac{\int_{\mathbb{R}^d} \boldsymbol{x} p_{\mathbf{y}|\mathbf{x}}(\boldsymbol{y}|\boldsymbol{x}) p_{\mathbf{x}}(\boldsymbol{x}) d\boldsymbol{x}}{p_{\mathbf{y}}(\boldsymbol{y})} \\
&= \frac{\frac{1}{(2\pi\sigma^2)^{\frac{d}{2}}} \int_{\mathbb{R}^d} \boldsymbol{x} \exp\{-\frac{1}{2\sigma^2}\|\boldsymbol{y} - \boldsymbol{x}\|^2\} p_{\mathbf{x}}(\boldsymbol{x}) d\boldsymbol{x}}{p_{\mathbf{y}}(\boldsymbol{y})}.
\end{aligned}
\tag{S14}
$$

Therefore, denoting the numerator by $q(\boldsymbol{y}) \triangleq \frac{1}{(2\pi\sigma^2)^{\frac{d}{2}}} \int_{\mathbb{R}^d} \boldsymbol{x} \exp\{-\frac{1}{2\sigma^2}\|\boldsymbol{y} - \boldsymbol{x}\|^2\} p_{\mathbf{x}}(\boldsymbol{x}) d\boldsymbol{x}$, we can write the Jacobian of $\boldsymbol{\mu}_1$ at $\boldsymbol{y}$ as

$$
\begin{aligned}
\frac{\partial \boldsymbol{\mu}(\boldsymbol{y})}{\partial \boldsymbol{y}} &= \frac{\frac{\partial q(\boldsymbol{y})}{\partial \boldsymbol{y}} p_{\mathbf{y}}(\boldsymbol{y}) - q(\boldsymbol{y}) (\nabla p_{\mathbf{y}}(\boldsymbol{y}))^\top}{p_{\mathbf{y}}^2(\boldsymbol{y})} \\
&= \frac{\frac{\partial q(\boldsymbol{y})}{\partial \boldsymbol{y}}}{p_{\mathbf{y}}(\boldsymbol{y})} - \frac{q(\boldsymbol{y})}{p_{\mathbf{y}}(\boldsymbol{y})} \frac{(\nabla p_{\mathbf{y}}(\boldsymbol{y}))^\top}{p_{\mathbf{y}}(\boldsymbol{y})} \\
&= \frac{\frac{\partial q(\boldsymbol{y})}{\partial \boldsymbol{y}}}{p_{\mathbf{y}}(\boldsymbol{y})} - \boldsymbol{\mu}_1(\boldsymbol{y}) \frac{(\nabla p_{\mathbf{y}}(\boldsymbol{y}))^\top}{p_{\mathbf{y}}(\boldsymbol{y})} \\
&= \frac{\frac{\partial q(\boldsymbol{y})}{\partial \boldsymbol{y}}}{p_{\mathbf{y}}(\boldsymbol{y})} - \boldsymbol{\mu}_1(\boldsymbol{y}) (\nabla \log p_{\mathbf{y}}(\boldsymbol{y}))^\top \\
&= \frac{\frac{\partial q(\boldsymbol{y})}{\partial \boldsymbol{y}}}{p_{\mathbf{y}}(\boldsymbol{y})} - \frac{1}{\sigma^2} \boldsymbol{\mu}_1(\boldsymbol{y}) (\boldsymbol{\mu}_1(\boldsymbol{y})^\top - \boldsymbol{y}^\top).
\end{aligned}
\tag{S15}
$$

Here, $\frac{\partial q(\boldsymbol{y})}{\partial \boldsymbol{y}} \in \mathbb{R}^{d \times d}$ denotes the Jacobian of $q : \mathbb{R}^d \to \mathbb{R}^d$ at $\boldsymbol{y}$, and we used the fact that $\nabla \log p_{\mathbf{y}}(\boldsymbol{y}) = \frac{1}{\sigma^2}(\boldsymbol{\mu}_1(\boldsymbol{y}) - \boldsymbol{y})$ (Efron, 2011; Miyasawa et al., 1961; Stein, 1981). The first term in (S15) can be further simplified as

$$
\begin{aligned}
\frac{\frac{\partial q(\boldsymbol{y})}{\partial \boldsymbol{y}}}{p_{\mathbf{y}}(\boldsymbol{y})} &= \frac{\frac{1}{(2\pi\sigma^2)^{\frac{d}{2}}} \int_{\mathbb{R}^d} \boldsymbol{x} \exp\{-\frac{1}{2\sigma^2}\|\boldsymbol{y} - \boldsymbol{x}\|^2\} \frac{1}{\sigma^2}(\boldsymbol{x} - \boldsymbol{y})^\top p_{\mathbf{x}}(\boldsymbol{x}) d\boldsymbol{x}}{p_{\mathbf{y}}(\boldsymbol{y})} \\
&= \frac{\int_{\mathbb{R}^d} \frac{1}{\sigma^2} \boldsymbol{x} (\boldsymbol{x} - \boldsymbol{y})^\top p_{\mathbf{y}|\mathbf{x}}(\boldsymbol{x}|\boldsymbol{y}) p_{\mathbf{x}}(\boldsymbol{x}) d\boldsymbol{x}}{p_{\mathbf{y}}(\boldsymbol{y})} \\
&= \int_{\mathbb{R}^d} \frac{1}{\sigma^2} \boldsymbol{x} (\boldsymbol{x} - \boldsymbol{y})^\top p_{\mathbf{x}|\mathbf{y}}(\boldsymbol{x}|\boldsymbol{y}) d\boldsymbol{x} \\
&= \frac{1}{\sigma^2} \left( \mathbb{E}[\mathbf{x}\mathbf{x}^\top|\mathbf{y} = \boldsymbol{y}] - \mathbb{E}[\mathbf{x}|\mathbf{y} = \boldsymbol{y}] \boldsymbol{y}^\top \right) \\
&= \frac{1}{\sigma^2} \left( \mathbb{E}[\mathbf{x}\mathbf{x}^\top|\mathbf{y} = \boldsymbol{y}] - \boldsymbol{\mu}_1(\boldsymbol{y}) \boldsymbol{y}^\top \right).
\end{aligned}
\tag{S16}
$$

Substituting (S16) back into (S15), we obtain

$$
\begin{aligned}
\frac{\partial \boldsymbol{\mu}_1(\boldsymbol{y})}{\partial \boldsymbol{y}} &= \frac{1}{\sigma^2}\left(\mathbb{E}[\mathbf{x}\mathbf{x}^\top|\mathbf{y}=\boldsymbol{y}] - \boldsymbol{\mu}_1(\boldsymbol{y})\,\boldsymbol{y}^\top\right) - \frac{1}{\sigma^2}\boldsymbol{\mu}_1(\boldsymbol{y})\left(\boldsymbol{\mu}_1(\boldsymbol{y})^\top - \boldsymbol{y}^\top\right) \\
&= \frac{1}{\sigma^2}\left(\mathbb{E}[\mathbf{x}\mathbf{x}^\top|\mathbf{y}=\boldsymbol{y}] - \boldsymbol{\mu}_1(\boldsymbol{y})\boldsymbol{\mu}_1(\boldsymbol{y})^\top\right) \\
&= \frac{1}{\sigma^2}\left(\mathbb{E}[\mathbf{x}\mathbf{x}^\top|\mathbf{y}=\boldsymbol{y}] - \mathbb{E}[\mathbf{x}|\mathbf{y}=\boldsymbol{y}]\mathbb{E}[\mathbf{x}|\mathbf{y}=\boldsymbol{y}]^\top\right) \\
&= \frac{1}{\sigma^2}\mathrm{Cov}(\mathbf{x}|\mathbf{y}=\boldsymbol{y}) \\
&= \frac{1}{\sigma^2}\boldsymbol{\mu}_2(\boldsymbol{y}).
\end{aligned}
\tag{S17}
$$

This completes the proof for $k=1$.

We now move on to the cases $k=2$ and $k\geq 3$ (second and third lines in (10)). Element $(i_1,\ldots,i_k)$ of the posterior $k$th order central moment can be expressed as

$$
\begin{aligned}
[\boldsymbol{\mu}_k(\boldsymbol{y})]_{i_1,\ldots,i_k} &= \mathbb{E}\left[(\mathbf{x}_{i_1} - [\boldsymbol{\mu}_1(\boldsymbol{y})]_{i_1})\cdots(\mathbf{x}_{i_k} - [\boldsymbol{\mu}_1(\boldsymbol{y})]_{i_k})\,\big|\,\mathbf{y}=\boldsymbol{y}\right] \\
&= \frac{\frac{1}{(2\pi\sigma^2)^{\frac{d}{2}}}\int_{\mathbb{R}^d}(\boldsymbol{x}_{i_1} - [\mu_1(\boldsymbol{y})]_{i_1})\cdots(\boldsymbol{x}_{i_k} - [\mu_1(\boldsymbol{y})]_{i_k})\exp\{-\frac{1}{2\sigma^2}\|\boldsymbol{y}-\boldsymbol{x}\|^2\}p_{\mathbf{x}}(\boldsymbol{x})d\boldsymbol{x}}{p_{\mathbf{y}}(\boldsymbol{y})} \\
&= \frac{q(\boldsymbol{y})}{p_{\mathbf{y}}(\boldsymbol{y})},
\end{aligned}
\tag{S18}
$$

where $q(\boldsymbol{y}) \triangleq \frac{1}{(2\pi\sigma^2)^{\frac{d}{2}}}\int_{\mathbb{R}^d}(\boldsymbol{x}_{i_1} - [\mu_1(\boldsymbol{y})]_{i_1})\cdots(\boldsymbol{x}_{i_k} - [\mu_1(\boldsymbol{y})]_{i_k})\exp\{-\frac{1}{2\sigma^2}\|\boldsymbol{y}-\boldsymbol{x}\|^2\}p_{\mathbf{x}}(\boldsymbol{x})d\boldsymbol{x}$.
Therefore, for any $i_{k+1}\in\{1,\ldots,d\}$, the derivative of $[\boldsymbol{\mu}_k(\boldsymbol{y})]_{i_1,\ldots,i_k}$ with respect to $\boldsymbol{y}_{i_{k+1}}$ is given by

$$
\begin{aligned}
\frac{\partial[\boldsymbol{\mu}_k(\boldsymbol{y})]_{i_1,\ldots,i_k}}{\partial \boldsymbol{y}_{i_{k+1}}} &= \frac{\frac{\partial q(\boldsymbol{y})}{\partial \boldsymbol{y}_{i_{k+1}}}p_{\mathbf{y}}(\boldsymbol{y}) - q(\boldsymbol{y})\frac{\partial p_{\mathbf{y}}(\boldsymbol{y})}{\partial \boldsymbol{y}_{i_{k+1}}}}{p_{\mathbf{y}}^2(\boldsymbol{y})} \\
&= \frac{\frac{\partial q(\boldsymbol{y})}{\partial \boldsymbol{y}_{i_{k+1}}}}{p_{\mathbf{y}}(\boldsymbol{y})} - \frac{q(\boldsymbol{y})}{p_{\mathbf{y}}(\boldsymbol{y})}\frac{\frac{\partial p_{\mathbf{y}}(\boldsymbol{y})}{\partial \boldsymbol{y}_{i_{k+1}}}}{p_{\mathbf{y}}(\boldsymbol{y})} \\
&= \frac{\frac{\partial q(\boldsymbol{y})}{\partial \boldsymbol{y}_{i_{k+1}}}}{p_{\mathbf{y}}(\boldsymbol{y})} - [\boldsymbol{\mu}_k(\boldsymbol{y})]_{i_1,\ldots,i_k}\frac{\partial \log p_{\mathbf{y}}(\boldsymbol{y})}{\partial \boldsymbol{y}_{i_{k+1}}} \\
&= \frac{\frac{\partial q(\boldsymbol{y})}{\partial \boldsymbol{y}_{i_{k+1}}}}{p_{\mathbf{y}}(\boldsymbol{y})} - \frac{1}{\sigma^2}[\boldsymbol{\mu}_k(\boldsymbol{y})]_{i_1,\ldots,i_k}\left([\boldsymbol{\mu}_1(\boldsymbol{y})]_{i_{k+1}} - \boldsymbol{y}_{i_{k+1}}\right),
\end{aligned}
\tag{S19}
$$

where in the last line we used the fact that $\nabla \log p_{\mathbf{y}}(\boldsymbol{y}) = \frac{1}{\sigma^2}(\boldsymbol{\mu}_1(\boldsymbol{y}) - \boldsymbol{y})$ (Efron, 2011; Miyasawa et al., 1961; Stein, 1981). The first term here can be written as

$$
\begin{aligned}
\frac{\frac{\partial q(\boldsymbol{y})}{\partial \boldsymbol{y}_{i_{k+1}}}}{p_{\mathbf{y}}(\boldsymbol{y})} &= \frac{\frac{1}{(2\pi\sigma^2)^{\frac{d}{2}}}\int\frac{\partial}{\partial \boldsymbol{y}_{i_{k+1}}}\left[(\boldsymbol{x}_{i_1} - [\boldsymbol{\mu}_1(\boldsymbol{y})]_{i_1})\cdots(\boldsymbol{x}_{i_k} - [\boldsymbol{\mu}_1(\boldsymbol{y})]_{i_k})\exp\{-\frac{1}{2\sigma^2}\|\boldsymbol{y}-\boldsymbol{x}\|^2\}\right]p_{\mathbf{x}}(\boldsymbol{x})d\boldsymbol{x}}{p_{\mathbf{y}}(\boldsymbol{y})} \\
&= \frac{\int -\frac{\partial[\boldsymbol{\mu}_1(\boldsymbol{y})]_{i_1}}{\partial \boldsymbol{y}_{i_{k+1}}}(\boldsymbol{x}_{i_2} - [\boldsymbol{\mu}_1(\boldsymbol{y})]_{i_2})\cdots(\boldsymbol{x}_{i_k} - [\boldsymbol{\mu}_1(\boldsymbol{y})]_{i_k})\exp\{-\frac{1}{2\sigma^2}\|\boldsymbol{y}-\boldsymbol{x}\|^2\}p_{\mathbf{x}}(\boldsymbol{x})d\boldsymbol{x}}{(2\pi\sigma^2)^{\frac{d}{2}}p_{\mathbf{y}}(\boldsymbol{y})} + \cdots \\
&\quad + \frac{\int -(\boldsymbol{x}_{i_1} - [\boldsymbol{\mu}_1(\boldsymbol{y})]_{i_1})\cdots(\boldsymbol{x}_{i_{k-1}} - [\boldsymbol{\mu}_1(\boldsymbol{y})]_{i_{k-1}})\frac{\partial[\boldsymbol{\mu}_1(\boldsymbol{y})]_{i_k}}{\partial \boldsymbol{y}_{i_{k+1}}}\exp\{-\frac{1}{2\sigma^2}\|\boldsymbol{y}-\boldsymbol{x}\|^2\}p_{\mathbf{x}}(\boldsymbol{x})d\boldsymbol{x}}{(2\pi\sigma^2)^{\frac{d}{2}}p_{\mathbf{y}}(\boldsymbol{y})} \\
&\quad + \frac{\int(\boldsymbol{x}_{i_1} - [\boldsymbol{\mu}_1(\boldsymbol{y})]_{i_1})\cdots(\boldsymbol{x}_{i_k} - [\boldsymbol{\mu}_1(\boldsymbol{y})]_{i_k})\frac{1}{\sigma^2}(\boldsymbol{x}_{i_{k+1}} - \boldsymbol{y}_{i_{k+1}})\exp\{-\frac{1}{2\sigma^2}\|\boldsymbol{y}-\boldsymbol{x}\|^2\}p_{\mathbf{x}}(\boldsymbol{x})d\boldsymbol{x}}{(2\pi\sigma^2)^{\frac{d}{2}}p_{\mathbf{y}}(\boldsymbol{y})}.
\end{aligned}
\tag{S20}
$$

Let us treat the cases $k = 2$ and $k \geq 3$ separately. When $k = 2$, the above expression contains precisely three terms, but the first two vanish. Indeed, the first term reduces to $-\frac{\partial[\boldsymbol{\mu}_1(\boldsymbol{y})]_{i_1}}{\partial \boldsymbol{y}_{i_3}}\mathbb{E}[\mathbf{x}_{i_2} - [\boldsymbol{\mu}_1(\mathbf{y})]_{i_2}|\mathbf{y} = \boldsymbol{y}] = -\frac{\partial[\boldsymbol{\mu}_1(\boldsymbol{y})]_{i_1}}{\partial \boldsymbol{y}_{i_3}}([\boldsymbol{\mu}_1(\mathbf{y})]_{i_2} - [\boldsymbol{\mu}_1(\mathbf{y})]_{i_2}) = 0$ and the second term to $-\frac{\partial[\boldsymbol{\mu}_1(\boldsymbol{y})]_{i_2}}{\partial \boldsymbol{y}_{i_3}}\mathbb{E}[\mathbf{x}_{i_1} - [\boldsymbol{\mu}_1(\mathbf{y})]_{i_1}|\mathbf{y} = \boldsymbol{y}] = -\frac{\partial[\boldsymbol{\mu}_1(\boldsymbol{y})]_{i_1}}{\partial \boldsymbol{y}_{i_2}}([\boldsymbol{\mu}_1(\mathbf{y})]_{i_1} - [\boldsymbol{\mu}_1(\mathbf{y})]_{i_1}) = 0$. Therefore, when $k = 2$ we are left only with the last term, which simplifies to

$$\frac{\frac{\partial q(\boldsymbol{y})}{\partial \boldsymbol{y}_{i_3}}}{p_{\mathbf{y}}(\boldsymbol{y})} = \frac{1}{\sigma^2}\mathbb{E}\left[(\mathbf{x}_{i_1} - [\boldsymbol{\mu}_1(\mathbf{y})]_{i_1})(\mathbf{x}_{i_2} - [\boldsymbol{\mu}_1(\mathbf{y})]_{i_2})(\mathbf{x}_{i_3} - \boldsymbol{y}_{i_3}) \,\big|\, \mathbf{y} = \boldsymbol{y}\right]$$

$$= \frac{1}{\sigma^2}\mathbb{E}\left[(\mathbf{x}_{i_1} - [\boldsymbol{\mu}_1(\mathbf{y})]_{i_1})(\mathbf{x}_{i_2} - [\boldsymbol{\mu}_1(\mathbf{y})]_{i_2})\mathbf{x}_{i_3} \,\big|\, \mathbf{y} = \boldsymbol{y}\right] - \frac{1}{\sigma^2}[\boldsymbol{\mu}_2(\boldsymbol{y})]_{i_1, i_2}\boldsymbol{y}_{i_3}$$

$$= \frac{1}{\sigma^2}\mathbb{E}\left[(\mathbf{x}_{i_1} - [\boldsymbol{\mu}_1(\mathbf{y})]_{i_1})(\mathbf{x}_{i_2} - [\boldsymbol{\mu}_1(\mathbf{y})]_{i_2})(\mathbf{x}_{i_3} - [\boldsymbol{\mu}_1(\mathbf{y})]_{i_3} + [\boldsymbol{\mu}_1(\mathbf{y})]_{i_3}) \,\big|\, \mathbf{y} = \boldsymbol{y}\right]$$

$$\quad - \frac{1}{\sigma^2}[\boldsymbol{\mu}_2(\boldsymbol{y})]_{i_1, i_2}\boldsymbol{y}_{i_3}$$

$$= \frac{1}{\sigma^2}[\boldsymbol{\mu}_3(\boldsymbol{y})]_{i_1, i_2, i_3} + \frac{1}{\sigma^2}[\boldsymbol{\mu}_2(\boldsymbol{y})]_{i_1, i_2}[\boldsymbol{\mu}_1(\mathbf{y})]_{i_3} - \frac{1}{\sigma^2}[\boldsymbol{\mu}_2(\boldsymbol{y})]_{i_1, i_2}\boldsymbol{y}_{i_3} \tag{S21}$$

Substituting this back into (S19), we obtain

$$\frac{\partial[\boldsymbol{\mu}_k(\boldsymbol{y})]_{i_1, i_2}}{\partial \boldsymbol{y}_{i_{k+1}}} = \frac{1}{\sigma^2}[\boldsymbol{\mu}_3(\boldsymbol{y})]_{i_1, i_2, i_3} + \frac{1}{\sigma^2}[\boldsymbol{\mu}_2(\boldsymbol{y})]_{i_1, i_2}[\boldsymbol{\mu}_1(\mathbf{y})]_{i_3} - \frac{1}{\sigma^2}[\boldsymbol{\mu}_2(\boldsymbol{y})]_{i_1, i_2}\boldsymbol{y}_{i_3}$$

$$\quad - \frac{1}{\sigma^2}[\boldsymbol{\mu}_2(\boldsymbol{y})]_{i_1, i_2}\left([\boldsymbol{\mu}_1(\boldsymbol{y})]_{i_3} - \boldsymbol{y}_{i_3}\right)$$

$$= \frac{1}{\sigma^2}[\boldsymbol{\mu}_3(\boldsymbol{y})]_{i_1, i_2, i_3}. \tag{S22}$$

This demonstrates that

$$[\boldsymbol{\mu}_3(\boldsymbol{y})]_{i_1, i_2, i_3} = \sigma^2\frac{\partial[\boldsymbol{\mu}_k(\boldsymbol{y})]_{i_1, i_2}}{\partial \boldsymbol{y}_{i_{k+1}}}, \tag{S23}$$

which completes the proof for $k = 2$.

When $k \geq 3$, none of the terms in (S20) vanish, and the expression reads

$$\frac{\frac{\partial q(\boldsymbol{y})}{\partial \boldsymbol{y}_{i_{k+1}}}}{p_{\mathbf{y}}(\boldsymbol{y})} = -\left([\boldsymbol{\mu}_{k-1}(\boldsymbol{y})]_{i_2, \ldots, i_k}\frac{\partial[\boldsymbol{\mu}_1(\boldsymbol{y})]_{i_1}}{\partial \boldsymbol{y}_{i_{k+1}}} + \cdots + [\boldsymbol{\mu}_{k-1}(\boldsymbol{y})]_{i_1, \ldots, i_{k-1}}\frac{\partial[\boldsymbol{\mu}_1(\boldsymbol{y})]_{i_k}}{\partial \boldsymbol{y}_{i_{k+1}}}\right)$$

$$\quad - \frac{1}{\sigma^2}[\boldsymbol{\mu}_k(\boldsymbol{y})]_{i_1, \ldots, i_k}\boldsymbol{y}_{i_{k+1}} + \frac{1}{\sigma^2}\mathbb{E}\left[(\mathbf{x}_{i_1} - [\boldsymbol{\mu}_1(\mathbf{y})]_{i_1})\cdots(\mathbf{x}_{i_k} - [\boldsymbol{\mu}_1(\mathbf{y})]_{i_k})\mathbf{x}_{i_{k+1}} \,\big|\, \mathbf{y} = \boldsymbol{y}\right]$$

$$= -\sum_{j=1}^{d}[\boldsymbol{\mu}_{k-1}(\boldsymbol{y})]_{\boldsymbol{\ell}_j}\frac{\partial[\boldsymbol{\mu}_1(\boldsymbol{y})]_{i_j}}{\partial \boldsymbol{y}_{i_{k+1}}} - \frac{1}{\sigma^2}[\boldsymbol{\mu}_k(\boldsymbol{y})]_{i_1, \ldots, i_k}\boldsymbol{y}_{i_{k+1}}$$

$$\quad + \frac{1}{\sigma^2}\mathbb{E}\left[(\mathbf{x}_{i_1} - [\boldsymbol{\mu}_1(\mathbf{y})]_{i_1})\cdots(\mathbf{x}_{i_k} - [\boldsymbol{\mu}_1(\mathbf{y})]_{i_k})(\mathbf{x}_{i_{k+1}} - [\boldsymbol{\mu}_1(\mathbf{y})]_{i_{k+1}} + [\boldsymbol{\mu}_1(\mathbf{y})]_{i_{k+1}}) \,\big|\, \mathbf{y} = \boldsymbol{y}\right]$$

$$= -\sum_{j=1}^{k}[\boldsymbol{\mu}_{k-1}(\boldsymbol{y})]_{\boldsymbol{\ell}_j}\frac{\partial[\boldsymbol{\mu}_1(\boldsymbol{y})]_{i_j}}{\partial \boldsymbol{y}_{i_{k+1}}} - \frac{1}{\sigma^2}[\boldsymbol{\mu}_k(\boldsymbol{y})]_{i_1, \ldots, i_k}\boldsymbol{y}_{i_{k+1}} + \frac{1}{\sigma^2}[\boldsymbol{\mu}_{k+1}(\boldsymbol{y})]_{i_1, \ldots, i_{k+1}}$$

$$\quad + \frac{1}{\sigma^2}[\boldsymbol{\mu}_k(\boldsymbol{y})]_{i_1, \ldots, i_k}[\boldsymbol{\mu}_1(\boldsymbol{y})]_{i_{k+1}}$$

$$= -\sum_{j=1}^{k}[\boldsymbol{\mu}_{k-1}(\boldsymbol{y})]_{\boldsymbol{\ell}_j}\frac{\partial[\boldsymbol{\mu}_1(\boldsymbol{y})]_{i_j}}{\partial \boldsymbol{y}_{i_{k+1}}} + \frac{1}{\sigma^2}[\boldsymbol{\mu}_{k+1}(\boldsymbol{y})]_{i_1, \ldots, i_{k+1}}$$

$$\quad + \frac{1}{\sigma^2}[\boldsymbol{\mu}_k(\boldsymbol{y})]_{i_1, \ldots, i_k}\left([\boldsymbol{\mu}_1(\boldsymbol{y})]_{i_{k+1}} - \boldsymbol{y}_{i_{k+1}}\right), \tag{S24}$$

where we used the definition $\boldsymbol{\ell}_j \triangleq (i_1, \ldots, i_{j-1}, i_{j+1} \ldots, i_k)$. Substituting this expression back into (S19), we obtain

$$\frac{\partial[\boldsymbol{\mu}_k(\boldsymbol{y})]_{i_1, \ldots, i_k}}{\partial \boldsymbol{y}_{i_{k+1}}} = -\sum_{j=1}^{k}[\boldsymbol{\mu}_{k-1}(\boldsymbol{y})]_{\boldsymbol{\ell}_j}\frac{\partial[\boldsymbol{\mu}_1(\boldsymbol{y})]_{i_j}}{\partial \boldsymbol{y}_{i_{k+1}}} + \frac{1}{\sigma^2}[\boldsymbol{\mu}_{k+1}(\boldsymbol{y})]_{i_1, \ldots, i_{k+1}}. \tag{S25}$$

This demonstrates that

$$
\begin{aligned}
[\boldsymbol{\mu}_{k+1}(\boldsymbol{y})]_{i_1,\dots,i_{k+1}} &= \sigma^2 \frac{\partial[\boldsymbol{\mu}_k(\boldsymbol{y})]_{i_1,\dots,i_k}}{\partial \boldsymbol{y}_{i_{k+1}}} + \sigma^2 \sum_{j=1}^{k} [\boldsymbol{\mu}_{k-1}(\boldsymbol{y})]_{\boldsymbol{\ell}_j} \frac{\partial[\boldsymbol{\mu}_1(\boldsymbol{y})]_{i_j}}{\partial \boldsymbol{y}_{i_{k+1}}} \\
&= \sigma^2 \frac{\partial[\boldsymbol{\mu}_k(\boldsymbol{y})]_{i_1,\dots,i_k}}{\partial \boldsymbol{y}_{i_{k+1}}} + \sum_{j=1}^{k} [\boldsymbol{\mu}_{k-1}(\boldsymbol{y})]_{\boldsymbol{\ell}_j} [\boldsymbol{\mu}_2(\boldsymbol{y})]_{i_j,i_{k+1}},
\end{aligned}
\tag{S26}
$$

where we used (S17). This completes the proof for $k \geq 3$.

## C  PROOF OF THEOREM 3

We will use the fact that for any $k \geq 1$, the posterior $k$th order central moment of $\boldsymbol{v}^\top \mathbf{x}$ can be written explicitly by expanding brackets as

$$
\begin{aligned}
\mathbb{E}\left[\left(\boldsymbol{v}^\top(\mathbf{x} - \boldsymbol{\mu}_1(\boldsymbol{y}))\right)^k \Big| \mathbf{y} = \boldsymbol{y}\right] &= \mathbb{E}\left[\left(\sum_{i=1}^d \boldsymbol{v}_i\, [\mathbf{x} - \boldsymbol{\mu}_1(\boldsymbol{y})]_i\right)^k \Bigg| \mathbf{y} = \boldsymbol{y}\right] \\
&= \sum_{i_1=1}^d \cdots \sum_{i_k=1}^d \boldsymbol{v}_{i_1}\dots\boldsymbol{v}_{i_k} \mathbb{E}\left[(\mathbf{x}_{i_1} - [\boldsymbol{\mu}_1(\boldsymbol{y})]_{i_1})\dots(\mathbf{x}_{i_1} - [\boldsymbol{\mu}_1(\boldsymbol{y})]_{i_k})|\mathbf{y} = \boldsymbol{y}\right] \\
&= \sum_{i_1=1}^d \cdots \sum_{i_k=1}^d \boldsymbol{v}_{i_1}\dots\boldsymbol{v}_{i_k} [\boldsymbol{\mu}_k(\boldsymbol{y})]_{i_1,\dots,i_k}.
\end{aligned}
\tag{S27}
$$

Let us start with the second moment. From (S27), it is given by

$$
\begin{aligned}
\mu_2^{\boldsymbol{v}}(\boldsymbol{y}) &= \sum_{i_1=1}^d \sum_{i_2=1}^d \boldsymbol{v}_{i_1} \boldsymbol{v}_{i_2} [\boldsymbol{\mu}_2(\boldsymbol{y})]_{i_1,i_2} \\
&= \boldsymbol{v}^\top \boldsymbol{\mu}_2(\boldsymbol{y}) \boldsymbol{v} \\
&= \sigma^2 \boldsymbol{v}^\top \frac{\partial \boldsymbol{\mu}_1(\boldsymbol{y})}{\partial \boldsymbol{y}} \boldsymbol{v} \\
&= \sigma^2 \nabla_{\boldsymbol{y}}\left(\boldsymbol{v}^\top \boldsymbol{\mu}_1(\boldsymbol{y})\right)^\top \boldsymbol{v} \\
&= \sigma^2 D_{\boldsymbol{v}}\left(\boldsymbol{v}^\top \boldsymbol{\mu}_1(\boldsymbol{y})\right) \\
&= \sigma^2 D_{\boldsymbol{v}} \mu_1^{\boldsymbol{v}}(\boldsymbol{y}).
\end{aligned}
\tag{S28}
$$

This proves the first line of (14).

Next, we derive the third moment. From (S27), it is given by

$$
\begin{aligned}
\mu_3^{\boldsymbol{v}}(\boldsymbol{y}) &= \sum_{i_1=1}^d \sum_{i_2=1}^d \sum_{i_3=1}^d \boldsymbol{v}_{i_1} \boldsymbol{v}_{i_2} \boldsymbol{v}_{i_3} [\boldsymbol{\mu}_3(\boldsymbol{y})]_{i_1,i_2,i_3} \\
&= \sigma^2 \sum_{i_1=1}^d \sum_{i_2=1}^d \sum_{i_3=1}^d \boldsymbol{v}_{i_1} \boldsymbol{v}_{i_2} \boldsymbol{v}_{i_3} \frac{\partial[\boldsymbol{\mu}_2(\boldsymbol{y})]_{i_1,i_2}}{\partial \boldsymbol{y}_{i_3}} \\
&= \sigma^2 \sum_{i_3=1}^d \boldsymbol{v}_{i_3} \frac{\partial\left(\boldsymbol{v}^\top \boldsymbol{\mu}_2(\boldsymbol{y})\boldsymbol{v}\right)}{\partial \boldsymbol{y}_{i_3}} \\
&= \sigma^2 \boldsymbol{v}^\top \nabla_{\boldsymbol{y}}\left(\boldsymbol{v}^\top \boldsymbol{\mu}_2(\boldsymbol{y})\boldsymbol{v}\right) \\
&= \sigma^2 D_{\boldsymbol{v}}\left(\boldsymbol{v}^\top \boldsymbol{\mu}_2(\boldsymbol{y})\boldsymbol{v}\right) \\
&= \sigma^2 D_{\boldsymbol{v}} \mu_2^{\boldsymbol{v}}(\boldsymbol{y}),
\end{aligned}
\tag{S29}
$$

where in the last line we used (S28). This proves the second line of (14).

Finally, we derive the $(k+1)$th moment for any $k \geq 3$. From (S27), it is given by

$$
\begin{aligned}
\boldsymbol{\mu}_{k+1}^{\boldsymbol{v}}(\boldsymbol{y}) &= \sum_{i_1=1}^{d} \cdots \sum_{i_{k+1}=1}^{d} \boldsymbol{v}_{i_1} \ldots \boldsymbol{v}_{i_{k+1}} [\boldsymbol{\mu}_{k+1}(\boldsymbol{y})]_{i_1,\ldots,i_{k+1}} \\
&= \sum_{i_1=1}^{d} \cdots \sum_{i_{k+1}=1}^{d} \boldsymbol{v}_{i_1} \ldots \boldsymbol{v}_{i_{k+1}} \left( \sigma^2 \frac{\partial [\boldsymbol{\mu}_k(\boldsymbol{y})]_{i_1,\ldots,i_k}}{\partial \boldsymbol{y}_{i_{k+1}}} + \sum_{j=1}^{k} [\boldsymbol{\mu}_{k-1}(\boldsymbol{y})]_{\boldsymbol{\ell}_j} [\boldsymbol{\mu}_2(\boldsymbol{y})]_{i_j,i_{k+1}} \right) \\
&= \sigma^2 \sum_{i_{k+1}=1}^{d} \boldsymbol{v}_{i_{k+1}} \frac{\partial}{\partial \boldsymbol{y}_{i_{k+1}}} \left( \sum_{i_1=1}^{d} \cdots \sum_{i_k=1}^{d} \boldsymbol{v}_{i_1} \ldots \boldsymbol{v}_{i_k} [\boldsymbol{\mu}_k(\boldsymbol{y})]_{i_1,\ldots,i_k} \right) + \\
&\quad \sum_{j=1}^{k} \left( \sum_{i_1=1}^{d} \cdots \sum_{i_{j-1}=1}^{d} \sum_{i_{j+1}=1}^{d} \cdots \sum_{i_{k+1}=1}^{d} \boldsymbol{v}_{i_1} \ldots \boldsymbol{v}_{i_{j-1}} \boldsymbol{v}_{i_{j+1}} \ldots \boldsymbol{v}_{i_k} [\boldsymbol{\mu}_{k-1}(\boldsymbol{y})]_{\boldsymbol{\ell}_j} \sum_{i_j=1}^{d} \sum_{i_{k+1}=1}^{d} \boldsymbol{v}_j \boldsymbol{v}_{i_{k+1}} [\boldsymbol{\mu}_2(\boldsymbol{y})]_{i_j,i_{k+1}} \right) \\
&= \sigma^2 \sum_{i_{k+1}=1}^{d} \boldsymbol{v}_{i_{k+1}} \frac{\partial \mu_k^{\boldsymbol{v}}(\boldsymbol{y})}{\partial \boldsymbol{y}_{i_{k+1}}} + \sum_{j=1}^{k} \mu_{k-1}^{\boldsymbol{v}}(\boldsymbol{y}) \mu_2^{\boldsymbol{v}}(\boldsymbol{y}) \\
&= \sigma^2 \boldsymbol{v}^{\top} \nabla_{\boldsymbol{y}} \mu_k^{\boldsymbol{v}}(\boldsymbol{y}) + k \mu_{k-1}^{\boldsymbol{v}}(\boldsymbol{y}) \mu_2^{\boldsymbol{v}}(\boldsymbol{y}) \\
&= \sigma^2 D_{\boldsymbol{v}} \mu_k^{\boldsymbol{v}}(\boldsymbol{y}) + k \mu_{k-1}^{\boldsymbol{v}}(\boldsymbol{y}) \mu_2^{\boldsymbol{v}}(\boldsymbol{y}),
\end{aligned}
\tag{S30}
$$

where in the second line we used (10). This completes the proof of the third line of (14).

## D   RELATED WORK: ESTIMATION OF HIGHER ORDER SCORES BY DENOISING

The work most related to ours is that of Meng et al. (2021). Here, we present their results while translating to our notation. Given a probability density $p_{\mathbf{y}}$ over $\mathbb{R}^d$, they define the $k$th order score $\mathbf{s}_k(\boldsymbol{y})$ as the tensor whose entry at multi-index $(i_1, i_2, ..., i_k)$ is

$$
[\mathbf{s}_k(\boldsymbol{y})]_{i_1,i_2,\ldots,i_k} \triangleq \frac{\partial^k}{\partial \boldsymbol{y}_{i_1} \partial \boldsymbol{y}_{i_2} \ldots \partial \boldsymbol{y}_{i_k}} \log p_{\mathbf{y}}(\boldsymbol{y}),
\tag{S31}
$$

for every $i_1, \ldots, i_k \in \{1, \ldots, d\}^k$. Using our notation, and under the assumption (5) that $\mathbf{y}$ is a noisy version of $\mathbf{x} \sim p_{\mathbf{x}}$, the denoising score matching method estimates the first-order score $\mathbf{s}_1(\boldsymbol{y})$, which is simply the gradient of the log-probability, $\nabla_{\mathbf{y}} \log p_{\mathbf{y}}(\boldsymbol{y})$. This is done by using Tweedie's formula, which links $\mathbf{s}_1$ with the first posterior moment (the MSE-optimal denoiser) as

$$
\boldsymbol{\mu}_1(\boldsymbol{y}) = \mathbb{E}[\mathbf{x}|\mathbf{y} = \boldsymbol{y}] = \boldsymbol{y} + \sigma^2 \mathbf{s}_1(\boldsymbol{y}).
\tag{S32}
$$

As noted by Meng et al. (2021), a similar relation links the second-order score with the second posterior moment (*i.e.*, the posterior covariance) as

$$
\boldsymbol{\mu}_2(\boldsymbol{y}) = \mathrm{Cov}(\mathbf{x}|\mathbf{y} = \boldsymbol{y}) = \sigma^4 \mathbf{s}_2(\boldsymbol{y}) + \sigma^2 I.
\tag{S33}
$$

Note from (S31) that $\mathbf{s}_2(\boldsymbol{y})$ is the Hessian of the log-probability, $\nabla_{\mathbf{y}}^2 \log p_{\mathbf{y}}(\boldsymbol{y})$, or equivalently the Jacobian of the gradient of the log probability, $\frac{\partial}{\partial \boldsymbol{y}} \nabla_{\mathbf{y}} \log p_{\mathbf{y}}(\boldsymbol{y})$. And since we have from (S32) that $\nabla_{\boldsymbol{y}} \log p_{\mathbf{y}}(\boldsymbol{y}) = \frac{1}{\sigma^2}(\boldsymbol{\mu}_1(\boldsymbol{y}) - \boldsymbol{y})$, Eq. (S33) can be equivalently written as

$$
[\boldsymbol{\mu}_2(\boldsymbol{y})]_{i_1,i_2} = \sigma^4 \frac{\partial}{\partial \boldsymbol{y}_{i_2}} \left[ \frac{\boldsymbol{\mu}_1(\boldsymbol{y}) - \boldsymbol{y}}{\sigma^2} \right]_{i_1} + \sigma^2 I = \sigma^2 \frac{\partial [\boldsymbol{\mu}_1(\boldsymbol{y})]_{i_1}}{\partial \boldsymbol{y}_{i_2}}.
\tag{S34}
$$

This illustrates that the second-order formula of Meng et al. (2021) is equivalent to (10).

Moving on to higher-order moments, following our notations, Lemma 1 of Meng et al. (2021) states that

$$
\mathbb{E}[\otimes^{k+1}\mathbf{x}|\mathbf{y} = \boldsymbol{y}] = \sigma^2 \frac{\partial}{\partial \boldsymbol{y}} \mathbb{E}[\otimes^k \mathbf{x}|\mathbf{y} = \boldsymbol{y}] + \sigma^2 \mathbb{E}[\otimes^k \mathbf{x}|\mathbf{y} = \boldsymbol{y}] \otimes \left( \mathbf{s}_1(\boldsymbol{y}) + \frac{\boldsymbol{y}}{\sigma^2} \right), \quad \forall k \geq 1, \tag{S35}
$$

where $\otimes^{k+1}\mathbf{x} \in \mathbb{R}^{d^k}$ denotes $k$-fold tensor multiplication. This lemma is used in Theorem 3 of Meng et al. (2021), to derive a recursion relating higher-order moments and scores. Substituting (S32), this relation can be written as

$$\mathbb{E}[\otimes^{k+1}\mathbf{x}|\mathbf{y} = \boldsymbol{y}] = \sigma^2 \frac{\partial}{\partial \boldsymbol{y}} \mathbb{E}[\otimes^k\mathbf{x}|\mathbf{y} = \boldsymbol{y}] + \mathbb{E}[\otimes^k\mathbf{x}|\mathbf{y} = \boldsymbol{y}] \otimes \boldsymbol{\mu}_1(\boldsymbol{y}), \quad \forall k \geq 1. \tag{S36}$$

Denoting the non-central posterior moment of order $k$ by $\boldsymbol{m}_k(\boldsymbol{y})$, Eq. (S36) can be written compactly as

$$\boldsymbol{m}_{k+1}(\boldsymbol{y}) = \sigma^2 \frac{\partial}{\partial \boldsymbol{y}} \boldsymbol{m}_k(\boldsymbol{y}) + \boldsymbol{m}_k(\boldsymbol{y}) \otimes \boldsymbol{\mu}_1(\boldsymbol{y}), \quad \forall k \geq 1. \tag{S37}$$

Writing out the elements of $\boldsymbol{m}_{k+1}(\boldsymbol{y})$ explicitly, this relation reads

$$[\boldsymbol{m}_{k+1}(\boldsymbol{y})]_{i_1,\ldots,i_{k+1}} = \sigma^2 \frac{\partial [\boldsymbol{m}_k(\boldsymbol{y})]_{i_1,\ldots,i_k}}{\partial \boldsymbol{y}_{i_{k+1}}} + [\boldsymbol{m}_k(\boldsymbol{y})]_{i_1,\ldots,i_k}[\boldsymbol{\mu}_1(\boldsymbol{y})]_{i_{k+1}}, \quad \forall k \geq 1. \tag{S38}$$

It is interesting to compare this expression with the recursion for the central moments in Theorem 2. We see that the non-central moments satisfy a sort of one-step recursion (if we disregard the dependence on $\boldsymbol{\mu}_1$), in the sense that $\boldsymbol{m}_{k+1}$ depends only on $\boldsymbol{m}_k$. In contrast, as can be seen in Theorem 2, the central moments satisfy a sort of two-step recursion (if we disregard the dependence on $\boldsymbol{\mu}_2$), in the sense that $\boldsymbol{\mu}_{k+1}(\boldsymbol{y})$ depends on both $\boldsymbol{\mu}_k(\boldsymbol{y})$ and $\boldsymbol{\mu}_{k-1}(\boldsymbol{y})$.

## E    POSTERIOR DISTRIBUTION FOR A GAUSSIAN MIXTURE PRIOR

In Fig. 1 and Fig. 3, we demonstrated our approach on one-dimensional and two-dimensional Gaussian mixtures, respectively. In both cases, we showed plots of the marginal posterior distribution in the direction of the first posterior principal component, as well as the posterior mean for a particular noisy input sample. Those simulations relied on the closed-form expressions of the posterior distribution and the marginal posterior distribution along some direction for a Gaussian mixture prior. In addition, Fig. 1 and Fig. 3 also contain the maximum entropy distribution estimated using our method, which uses the numerical derivatives of the posterior mean. Here as well we used the numerical derivatives of the posterior mean function, which we computed in closed-from. We now present these closed-form expressions for completeness.

Suppose $p_{\mathbf{x}}$ is a mixture of $L$ Gaussians,

$$p_{\mathbf{x}}(\boldsymbol{x}) = \sum_{\ell=1}^{L} \pi_\ell \mathcal{N}(\boldsymbol{x}; m_\ell, \Sigma_\ell). \tag{S39}$$

Let c be a random variable taking values in $\{1, \ldots, L\}$ with probabilities $\pi_1, \ldots, \pi_L$. Then we can think of $\mathbf{x}$ as drawn from the $\ell$th Gaussian conditioned on the event that c $= \ell$. Therefore,

$$\begin{aligned} p_{\mathbf{x}|\mathbf{y}}(\boldsymbol{x}|\boldsymbol{y}) &= \sum_{\ell=1}^{L} p_{\mathbf{x}|\mathbf{y},\mathbf{c}}(\boldsymbol{x}|\boldsymbol{y}, \ell) p_{\mathbf{c}|\mathbf{y}}(\ell|\boldsymbol{y}) \\ &= \sum_{\ell=1}^{L} p_{\mathbf{x}|\mathbf{y},\mathbf{c}}(\boldsymbol{x}|\boldsymbol{y}, \ell) \frac{p_{\mathbf{y}|\mathbf{c}}(\boldsymbol{y}|\ell) p_{\mathbf{c}}(\ell)}{p_{\mathbf{y}}(\boldsymbol{y})} \\ &= \sum_{\ell=1}^{L} \mathcal{N}(\boldsymbol{x}; \bar{\boldsymbol{m}}_\ell, \bar{\Sigma}_\ell) \frac{\rho_\ell \pi_\ell}{\sum_{\ell'=1}^{L} \rho_{\ell'} \pi_{\ell'}}, \end{aligned} \tag{S40}$$

where we denoted

$$\begin{aligned} \rho_i &= \mathcal{N}(\boldsymbol{y}; \boldsymbol{m}_i, \Sigma_i + \sigma^2 \boldsymbol{I}), \\ \bar{\boldsymbol{m}}_i &= \Sigma_i (\Sigma_i + \sigma^2 \boldsymbol{I})^{-1}(\boldsymbol{y} - \boldsymbol{m}_i) + \boldsymbol{m}_i, \\ \bar{\Sigma}_i &= \Sigma_i - \Sigma_i (\Sigma_i + \sigma^2 \boldsymbol{I})^{-1} \Sigma_i. \end{aligned} \tag{S41}$$

As for the marginal posterior distribution along some direction $\boldsymbol{v}$, it is easy to show that

$$
\begin{aligned}
p_{\boldsymbol{v}^\top \mathbf{x}|\mathbf{y}}(\alpha|\boldsymbol{y}) &= \sum_{\ell=1}^{L} p_{\boldsymbol{v}^\top \mathbf{x}|\mathbf{y},c}(\alpha|\boldsymbol{y},\ell) p_{c|\mathbf{y}}(\ell|\boldsymbol{y}) \\
&= \sum_{\ell=1}^{L} p_{\boldsymbol{v}^\top \mathbf{x}|\mathbf{y},c}(\alpha|\boldsymbol{y},\ell) \frac{p_{\mathbf{y}|c}(\boldsymbol{y}|\ell) p_c(\ell)}{p_{\mathbf{y}}(\boldsymbol{y})} \\
&= \sum_{\ell=1}^{L} \mathcal{N}(\alpha; \boldsymbol{v}^\top \bar{\boldsymbol{m}}_\ell, \boldsymbol{v}^\top \bar{\Sigma}_\ell \boldsymbol{v}) \frac{\rho_\ell \pi_\ell}{\sum_{\ell'=1}^{L} \rho_{\ell'} \pi_{\ell'}}.
\end{aligned}
\tag{S42}
$$

## F  PROOF OF COROLLARY 1

We start by reminding the reader of (4) :

$$
\begin{aligned}
\mu_2(y) &= \sigma^2 \, \mu_1'(y), \\
\mu_3(y) &= \sigma^2 \, \mu_2'(y), \\
\mu_{k+1}(y) &= \sigma^2 \, \mu_k'(y) + k\mu_{k-1}(y)\mu_2(y), \quad k \geq 3.
\end{aligned}
$$

We will prove by complete induction that

$$
\mu_k^{(m)} = 0 \quad \text{for all } k \geq 2 \text{ and } m \geq 1.
\tag{S43}
$$

**Base**   Note that since for any $m \geq 2$ we have $\mu_1^{(m)}(y^*) = 0$, for any $m \geq 1$ we have

$$
\begin{aligned}
\mu_2^{(m)}(y^*) &= \sigma^2 \mu_1^{(m+1)}(y^*) \\
&= 0 \\
\mu_3^{(m)}(y^*) &= \sigma^2 \mu_2^{(m+1)}(y^*) \\
&= \sigma^4 \mu_1^{(m+2)}(y^*) \\
&= 0 \\
\mu_4^{(m)}(y^*) &= \sigma^2 \mu_3^{(m+1)}(y^*) + 3 \left. \frac{\partial^m}{\partial y^m} \left( \mu_2^2(y) \right) \right|_{y=y^*} \\
&\overset{(1)}{=} \sigma^2 \mu_3^{(m+1)}(y^*) + 3 \sum_{l=0}^{m} \binom{m}{l} \mu_2^{(m-l)}(y^*) \mu_2^{(l)}(y^*) \\
&= \sigma^2 \mu_3^{(m+1)}(y^*) + 3 \left( \mu_2^{(m)}(y^*)\mu_2(y^*) + \cdots + \mu_2(y^*)\mu_2^{(m)}(y^*) \right) \\
&= \sigma^2 \mu_3^{(m+1)}(y^*) \\
&= 0,
\end{aligned}
\tag{S44}
$$

where (1) results from the general Leibniz rule.

**Induction**   Assume that $\mu_n^{(m)}(y^*) = 0$ for all $4 \leq n < k+1$ and $m \geq 1$. Then,

$$
\begin{aligned}
\mu_{k+1}^{(m)}(y^*) &= \left. \frac{\partial^m}{\partial y^m} \left( \sigma^2 \, \mu_k'(y) + k\mu_{k-1}(y)\mu_2(y) \right) \right|_{y=y*} \\
&= \sigma^2 \, \mu_k^{(m+1)}(y^*) + k \left. \frac{\partial^m}{\partial y^m} \left( \mu_{k-1}(y)\mu_2(y) \right) \right|_{y=y^*} \\
&\overset{(1)}{=} \sigma^2 \, \mu_k^{(m+1)}(y^*) + k \sum_{l=0}^{m} \binom{m}{l} \mu_{k-1}^{(m-l)}(y^*) \mu_2^{(l)}(y^*) \\
&= \sigma^2 \, \mu_k^{(m+1)}(y^*) + k\mu_{k-1}^{(m)}(y^*)\mu_2(y^*) + ... + k\mu_{k-1}(y^*)\mu_2^{(m)}(y^*) \\
&\overset{(2)}{=} 0,
\end{aligned}
\tag{S45}
$$

where for (1) the general Leibniz rule was used again, and in (2) we used our induction assumption. This concludes the induction.

Using (S43) we therefore obtain for all $k \geq 3$ that

$$
\begin{aligned}
\mu_{k+1}(y^*) &= k\mu_{k-1}(y^*)\mu_2(y^*), \\
&= k(k-2)\mu_2^2(y^*)\mu_{k-3}(y^*) \\
&= k(k-2)(k-4)\mu_2^3(y^*)\mu_{k-5}(y^*) \\
&= ... \\
&= \begin{cases} k!!\mu_2^{\frac{k+1}{2}}(y^*) & k \text{ is odd,} \\ 0 & k \text{ is even.} \end{cases}
\end{aligned}
\tag{S46}
$$

Since $\mu_3(y^*) = \sigma^2\mu_2(y^*) = 0$ as well, the posterior moments are the same as those of a Gaussian distribution. Indeed, the central moments of a random variable $z \sim \mathcal{N}(\mathbb{E}[z], \sigma^2)$ are given by

$$
\mathbb{E}[(z - \mathbb{E}[z])^d] = \begin{cases} \sigma^d(d-1)!! & d \text{ is even,} \\ 0 & d \text{ is odd.} \end{cases}
\tag{S47}
$$

To conclude the proof, all that remains to show is moment-determinacy (*i.e.*, that the sequence of moments uniquely determines the distribution). This is the case, since the moments of a Gaussian distribution are trivially verified to satisfy *e.g.*, Condition $(h6)$ of (Lin, 2017). This implies that the posterior is moment-determinate, and is Gaussian.

## G EXPERIMENTAL DETAILS

Algorithm 1 requires three hyper-parameters as input. The first is the small constant $c$, which is used for the linear approximation in (15). The second is $N$, which is the number of principal components we seek. The last is $K$, which is the number of iterations to preform. In all our experiments we used $c = 10^{-5}$ and $N = 3$. For the N2V experiments we used $K = 100$ while for the rest we used $K = 50$.

Figure S1 depicts the convergence of the subspace iteration method for two different domains. For each noisy image and patch for which we find the principal components (marked in red), the plot to the right shows the convergence of the first $N = 3$ principal components. Specifically, for each principal component $v_i$, we calculate its inner product with the same principal component in the previous iteration. As the graph shows, $K = 50$ iterations suffice for convergence.

## H THE IMPACT OF THE JACOBIAN-VECTOR DOT-PRODUCT LINEAR APPROXIMATION

As described in Sec 4.1, Alg. 1 calls for calculating the Jacobian-vector dot-product of the denoiser. While for neural denoisers this calculation can be done via automatic differentiation, we propose using a linear approximation instead (See Eq. (15)). This can reduce the computational burden, while retaining high-accuracy in the computed eigenvectors. For example, in an experiment using SwinIR and $\sigma = 50$, the cosine similarity between the principal components computed with the approximation and those computed with automatic differentiation typically reaches around 0.97 at the $50th$ iteration. However, in terms of computational burden, the differences can sometimes be dramatic. For example, with the SwinIR model, when calculating one eigenvector for a patch of size $80 \times 92$, the memory footprint using automatic differentiation reaches 12GB, while using the linear approximation method it only reaches 2GB. These differences will increase for running on larger images. A visual comparison of the resulting principal component can be found in Fig. S2.

## I THE IMPACT OF ESTIMATING $\sigma$

Our theoretical analysis is developed for non-blind denoising, and accordingly, most of our experiments conform to this setting. These include the experiments on faces (Fig. 2 and Fig. S7), on

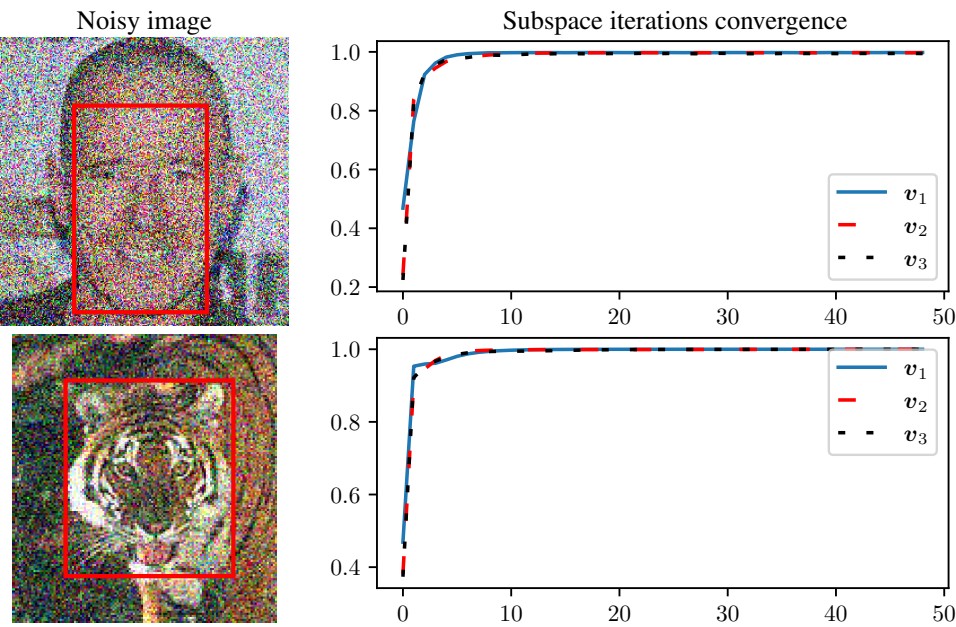

Figure S1: **Convergence of the subspace iteration method**. In each row one noisy image is shown with a red patch marking the region for which the posterior principal components are calculated. To the right, we plot for each of the first 3 principal components the inner product between the principal component in consecutive iterations. As the graph shows, $K = 50$ iterations suffice to guarantee convergence in those domains.

MNIST digits (Fig. 4), on natural images (top part of Fig. 5, S5 and S6), and the toy problem of Fig. 3. Namely, in all those experiments the noise level $\sigma$ was assumed known.

Nevertheless, we show empirically that our method can also work well in the blind setting. This is the case in the real microscopy images (bottom part of Fig. 5). In this experiment, we estimated $\sigma$ using the naive formula $\hat{\sigma}^2 = \frac{1}{d}\|\boldsymbol{\mu}_1(\boldsymbol{y}) - \boldsymbol{y}\|^2$, where $\boldsymbol{\mu}_1(\boldsymbol{y})$ is the (blind) N2V denoiser. It is certainly possible to employ more advanced noise-level estimation methods in order to obtain an even more accurate estimate for $\sigma$. Indeed, noise-level estimation, particularly for white Gaussian noise, has been heavily researched, and as of today state-of-the-art methods reach very-high precision (Chen et al., 2015; Khmag et al., 2018; Kokil & Pratap, 2021; Liu & Lin, 2012; Liu et al., 2013). For example, when the real $\sigma$ equals 10, the error in estimating sigma is around 0.05 (see *e.g.*, Chen et al. (2015)). However, we find that even with the naive method described above, we get quite accurate results. Particularly, the impact of small inaccuracies in $\sigma$ on our uncertainty estimation turn out to be very small. To illustrate this, we applied our method with a SwinIR model that was trained for $\sigma = 50$, on images with noise levels of $\sigma = 47.5, 52.5$. This accounts for $5\%$ errors in $\sigma$, that are significantly higher than typical $0.5\%$ errors of good noise level estimation techniques. Despite the inaccuracies in $\sigma$, the eigenvectors produced using our method are quite similar, as can be seen in Fig. S3.

## J  USE IN NON-GAUSSIAN SETTINGS

In Sec. 5 we empirically show our method provides sensible results also on real microscopy images (bottom part of Fig. 5), where the noise model is not known. In this setting, the noise distribution in each pixel is likely close to Poisson-Gaussian, the noise level is unknown, and it is not even clear if the noise is completely white. However, the theory developed in this paper holds only for the non-blind Gaussian denoising case. We therefore aim to provide here intuition as to why our method can still find meaningful results for blind Gaussian denoising.

Suppose that the observation model is

$$\mathbf{y} = \mathbf{x} + \sigma\mathbf{n}, \tag{S48}$$

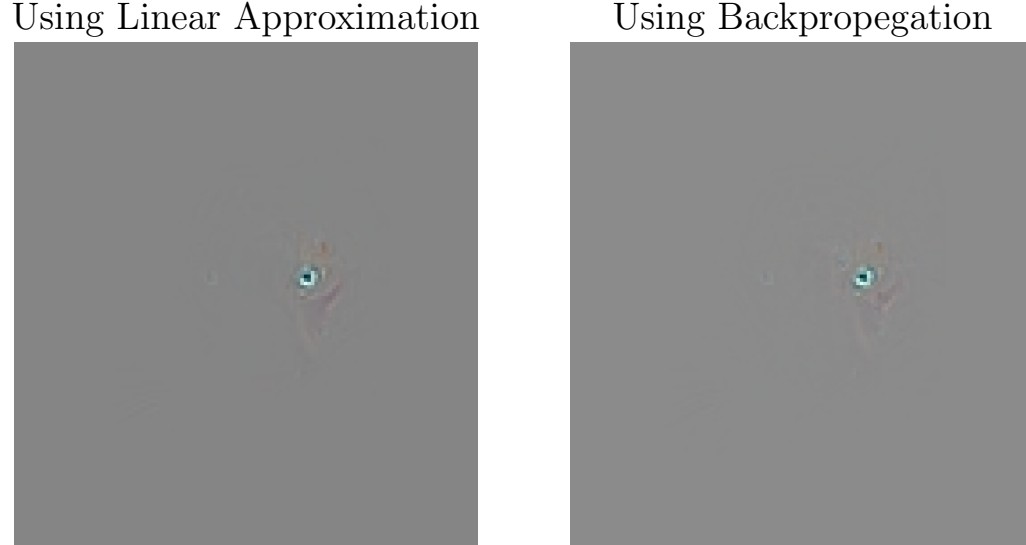

Figure S2: **The impact of the linear approximation on the calculated principal component.** The first principal component calculated with SwinIR and $\sigma = 50$, using the linear approximation in Eq. (15), and using automatic differentiation (Backpropegation). Both methods achieve similar results, with a cosine similarity of 0.96 over 50 iterations. However, the linear approximation methods uses drastically less memory.

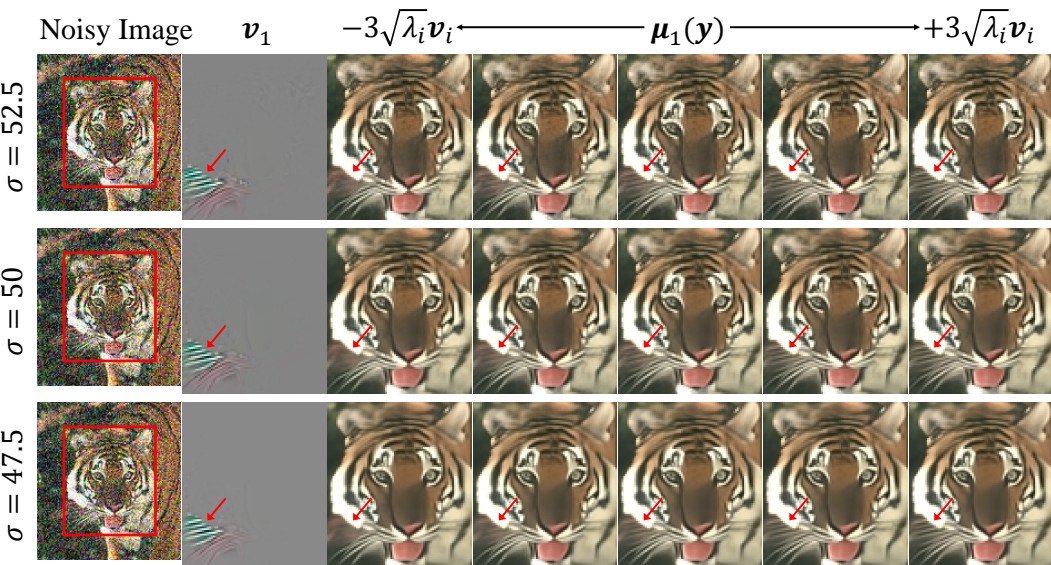

Figure S3: **The effect of small inaccuracies in $\sigma$ on uncertainty estimation.** The first principal component calculated using SwinIR, for an assumed $\sigma = 50$, for three different actual noise levels in the image.

where $\mathbf{x}$ is a random vector taking values in $\mathbb{R}^d$, the noise $\mathbf{n} \sim \mathcal{N}(0, \boldsymbol{I}_d)$ is a multivariate Gaussian vector that is statistically independent of $\mathbf{x}$, and the noise level $\sigma$ is a random variable sampled from some distribution $p_\sigma$. The noise level is unknown to the denoiser (all that is known is the distribution of noise levels $p_\sigma$).

In this case, the Jacobian of the MSE-optimal denoiser is given by

$$
\begin{aligned}
\frac{\partial \boldsymbol{\mu}_1(\boldsymbol{y})}{\partial \boldsymbol{y}} &= \frac{\partial \mathbb{E}[\mathbf{x}|\mathbf{y} = \boldsymbol{y}]}{\partial \boldsymbol{y}} = \\
&= \frac{\partial}{\partial \boldsymbol{y}} \mathbb{E}\left[\mathbb{E}[\mathbf{x}|\mathbf{y} = \boldsymbol{y}, \sigma = \sigma]|\mathbf{y} = \boldsymbol{y}\right] = \\
&= \mathbb{E}\left[\frac{\partial}{\partial \boldsymbol{y}} \mathbb{E}[\mathbf{x}|\mathbf{y} = \boldsymbol{y}, \sigma = \sigma]\Big|\mathbf{y} = \boldsymbol{y}\right] = \\
&= \mathbb{E}\left[\frac{\mathrm{Cov}(\mathbf{x}|\mathbf{y} = \boldsymbol{y}, \sigma)}{\sigma^2}\Big|\mathbf{y} = \boldsymbol{y}\right],
\end{aligned}
\tag{S49}
$$

where we used the law of total expectation in the second line, and Theorem 2 in the last line. Namely, instead of $\frac{\mathrm{Cov}(\mathbf{x}|\boldsymbol{y})}{\sigma^2}$, which we had in the Gaussian setting, here the Jacobian reveals the *mean* of the posterior covariance divided by $\sigma^2$, where the mean is taken over all possible noise levels $\sigma$. This matrix is a linear combination of the posterior covariances corresponding to different noise levels, so that it captures some notion of spread about the posterior mean, similarly to the regular posterior covariance that arises in the non-blind setting. Thus, intuitively, we expect that the top eigenvectors of this matrix capture meaningful uncertainty directions, similarly to the non-blind setting.

## K  VALIDATION OF THE PREDICTED PRINCIPAL COMPONENTS

It is impossible to *directly* measure the quality of our estimated posterior PCs, since denoising datasets contain only one clean image $\boldsymbol{x}$ for each noisy image $\boldsymbol{y}$. This single $\boldsymbol{x}$ is just one sample from the posterior $p_{\mathbf{x}|\mathbf{y}}$ and therefore it cannot be used to extract a ground-truth posterior covariance matrix or ground-truth PCs to compare against. To validate our method beyond the controlled toy-experiment of Fig. 3, in which the ground-truth posterior distribution was known analytically and thus so were the PCs, here we provide the two following experiments.

First, we employ the use of a diffusion-based posterior sampler for inverse-problems to generate many posterior samples for a noisy image $\boldsymbol{y}$. The posterior principal components can then by extracted by performing PCA on those samples. We note that this approach is impractical for real-world applications because of its very high computational cost, and is brought here only for evaluating our method against some baseline. Indeed, when using a diffusion-based posterior sampler, each sample requires many neural function evaluations (NFEs) to generate, and many samples are needed for obtaining accurate PCs. This is while our method can faithfully extract each posterior PC with only 10 NFEs, as shown in the convergence graphs in Fig. S1.

For each noisy image, we generated many posterior samples using DDNM (Wang et al., 2023) and used them to calculate the PCs of the posterior. As can be seen in Fig. S4, as the number of posterior samples increases, the PCs estimated using this baseline become cleaner and more similar to our PCs. However, even with 500 samples, the PCs of this baseline do not seem to have fully converged, and generating 500 posterior samples using DDNM requires 50,000 NFEs. Therefore, for extracting *e.g.*, 5 PCs, our method is roughly $1000\times$ faster than this naive approach.

We further supply quantitative results in Tab. 1, over 100 randomly selected images from CelebA-19 Baranchuk et al. (2022), a subset of CelebAMask-HQtest Lee et al. (2020). First, the empirical mean of the samples generated by the posterior samples should theoretically approximate the posterior mean, which is the MSE-optimal restoration. As we verify, this estimate is indeed very close to our denoiser's output, and they both achieve practically the same RMSE to the ground-truth images. Second, we compare the PCs of our method to those generated by the suggested baseline by measuring the norm of the error after projecting it onto these PCs. The larger this norm, the larger the portion of the error that these PCs account for. Mathematically, this measure is defined as $\|\boldsymbol{V}^T(\boldsymbol{x} - \boldsymbol{\mu}_1(\boldsymbol{y}))\|_2^2$, where $\boldsymbol{V}$ is a matrix containing the PCs as columns. We compute the ratio of this norm relative to the measured MSE, and find that the mean of this measure for both methods is also very close.

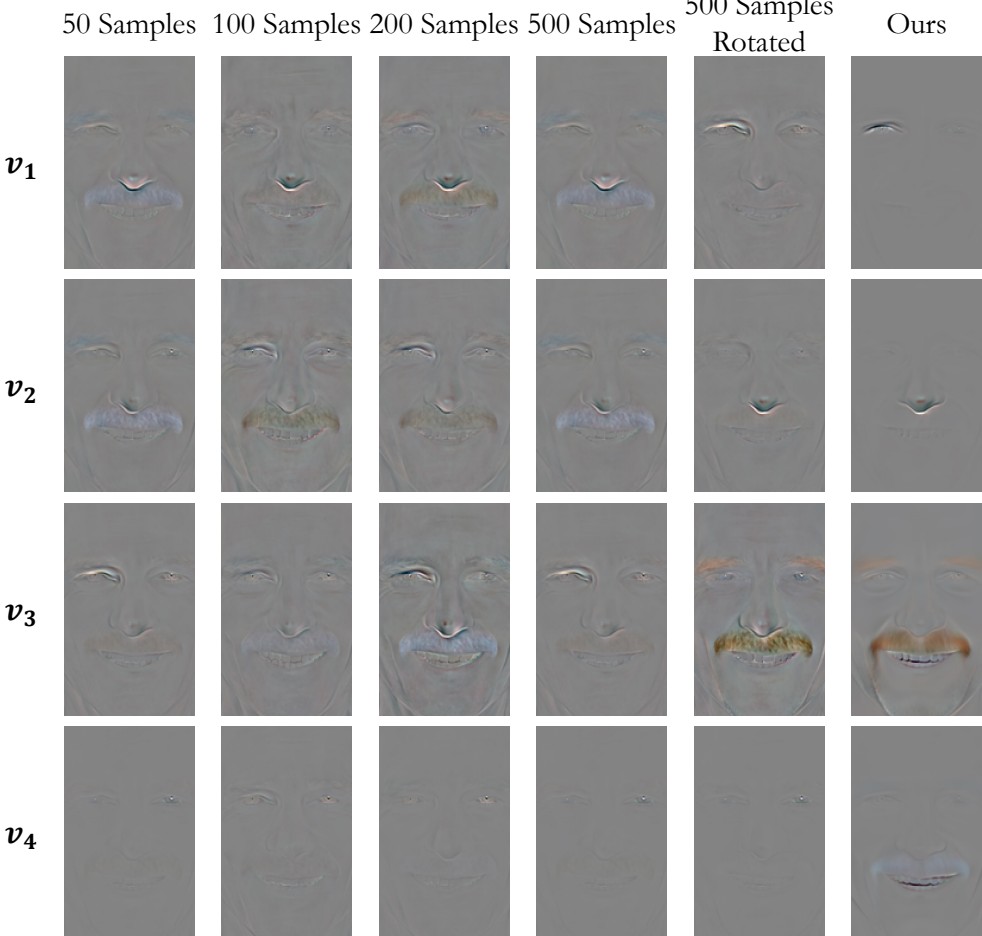

Figure S4: **Comparison to PCs computed by applying SVD on different numbers of posterior samples**. The posterior samples were produced using DDNM Wang et al. (2023). In the second column to the right, we rotate the produced PCs to best match our estimated PCs, while constraining them to remain orthonormal (using Procrustes analysis). The fact that the rotated PCs are quite similar to our PCs, shows that both sets of PCs span similar subspaces. However, as can be seen, our PCs are more disentangled within that subspace.

Table 1: Quantitative evaluation of the estimated PCs.

|  | RMSE$(\boldsymbol{x}, \boldsymbol{\mu}_1(\boldsymbol{y})) \downarrow$ | $\|\boldsymbol{V}^T(\boldsymbol{x} - \boldsymbol{\mu}_1(\boldsymbol{y}))\|_2^2/\text{MSE}(\boldsymbol{x}, \boldsymbol{\mu}_1(\boldsymbol{y})) \uparrow$ |
|---|---|---|
| **Baseline** | $4.026 \cdot 10^{-2}$ | $7.767 \cdot 10^{-3}$ |
| **Ours** | $4.022 \cdot 10^{-2}$ | $7.638 \cdot 10^{-3}$ |

Finally, we verify the predicted eigenvalues by comparing the projected test error over the first PC, $\boldsymbol{v}_1^T(\boldsymbol{x} - \boldsymbol{\mu}_1(\boldsymbol{y}))$, to the predicted 1st eigenvalue $\lambda_1$. The average of the ratio between those two quantities should theoretically be 1. For the same 100 randomly sampled face images, we found that the average of this ratio is 1.03. We also verified the predicted eigenvalues by calculating the ratio for the natural images domain. For this, we randomly selected 100 natural images from the CBSD (Martin et al., 2001), Kodak (Franzen, 1999) and McMaster (Zhang et al., 2011) datasets, and applied our algorithm using SwinIR (Liang et al., 2021). For each image we calculated the PCs on

Table 2: Quantitative evaluation of the estimated marginal posterior distributions.

|  | Face images NLL ↓ | Natural Images NLL ↓ |
|---|---|---|
| **Moments 1 & 2** | 1.83 | 0.05 |
| **Moments 1 − 4** | 1.81 | 0.03 |

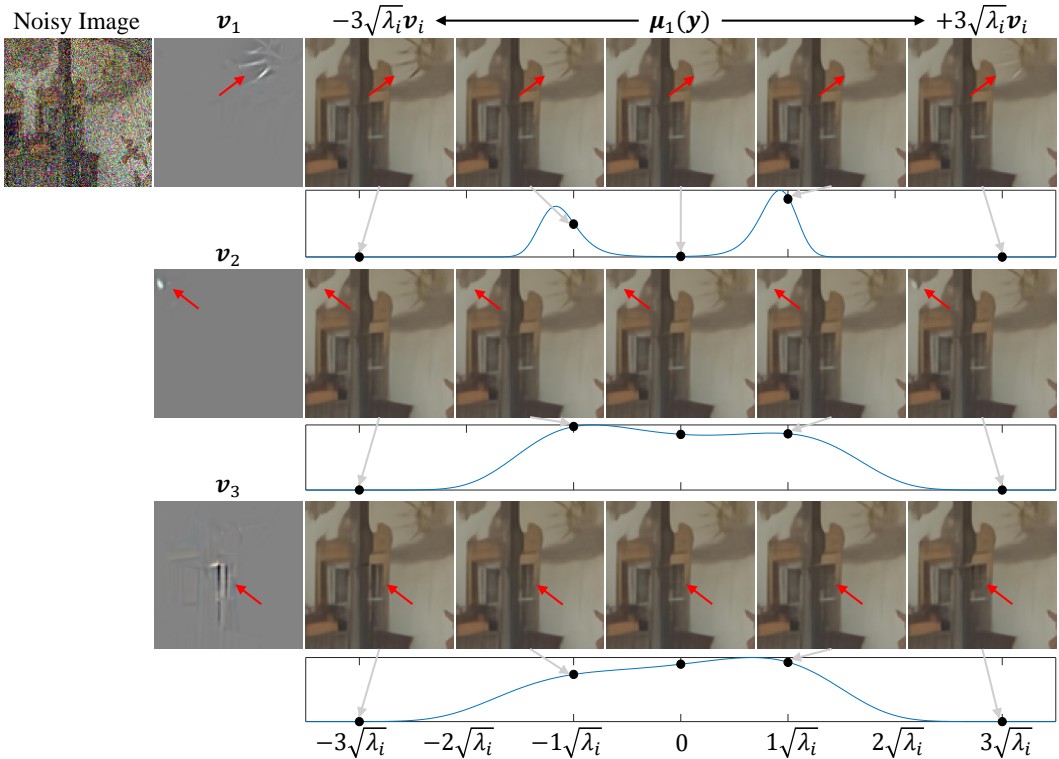

Figure S5: **Additional examples on natural images using SwinIR** (Liang et al., 2021). In each row, one of the first three PCs corresponding to the noisy image is shown on the left. On the right, images along the PC are shown above the marginal posterior distribution estimated for this direction. The principal components reveal uncertainty in delicate parts of the wall-painting, such as the thin rays of the sun, or the existence of mullions in the windows.

a $100 \times 100$ sized patch, located randomly within the image. For these images, the ratio computed was $0.93$.

In addition, to quantitatively verify that the marginal posterior distributions we estimate along the PCs are accurate, we measure the negative log likelihood (NLL) of the ground-truth images projected onto those directions (lower is better). We compared this to the NLL of a Gaussian distribution defined by only the first two estimated moments. Tab. 2 provides the results for the same 100 randomly selected face images and natural images. In both cases, the NLL of our estimation is lower.

## L  ADDITIONAL RESULTS

Figures S5 and S6 provide additional results on test images from the McMaster (Zhang et al., 2011) dataset and images from ImageNet (Deng et al., 2009). In the supplementary material we attach a video showing more examples on face images, demonstrating different semantic principal components.

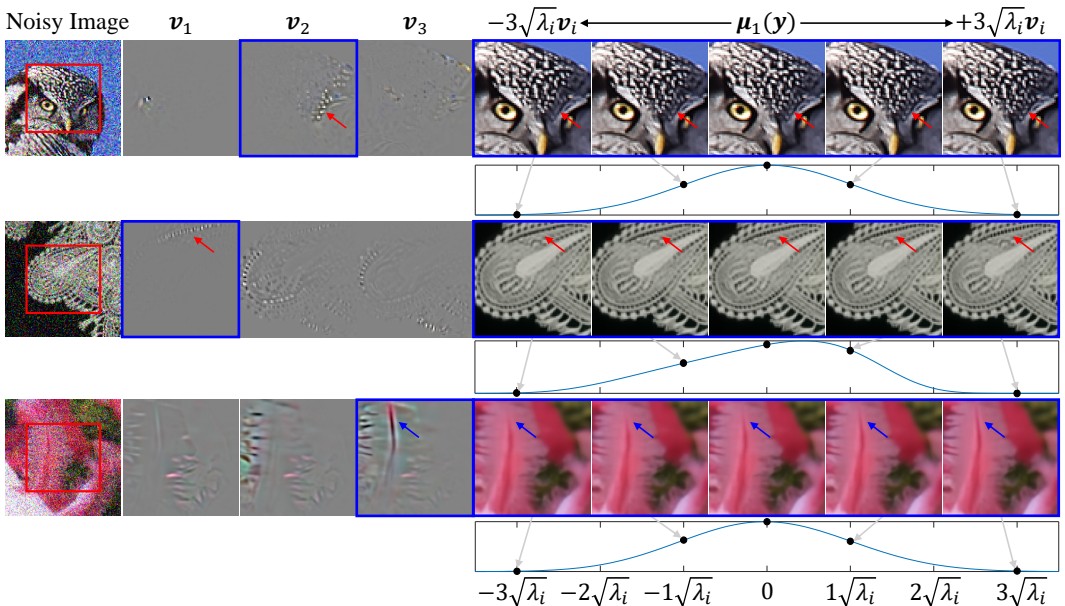

Figure S6: **Additional examples on natural images using SwinIR** (Liang et al., 2021). In each row, the first three PCs corresponding to the noisy image are shown on the left, and one is marked in blue. On the right, images along the marked PC are shown above the marginal posterior distribution estimated for this direction. The principal components catch semantic directions such as the pattern on the owl's feathers, the embroidery pattern, or the length of the Axolotl's gills.

## L.1 POLYNOMIAL FITTING EXAMPLES

As discussed briefly in Sec. 5, we experimented with fitting a polynomial to the function $f(\alpha) = \boldsymbol{v}^\top \boldsymbol{\mu}_1(\boldsymbol{y} + \alpha \boldsymbol{v})$, and using the derivatives of the polynomial at $\alpha = 0$ instead of using numerical derivatives of $f(\alpha)$ itself at $\alpha = 0$. Here, we provide the results of an experiment where we fit a polynomial of degree six over the range $\left[-\sqrt{\lambda_i}, \sqrt{\lambda_i}\right]$ for the $i$th principal component. As can be seen in Fig. S7, the marginal distribution estimates are quite smooth. Presumably, these posterior estimates are smoother than the true posterior, as the low degree polynomial smooths the directional posterior mean function.

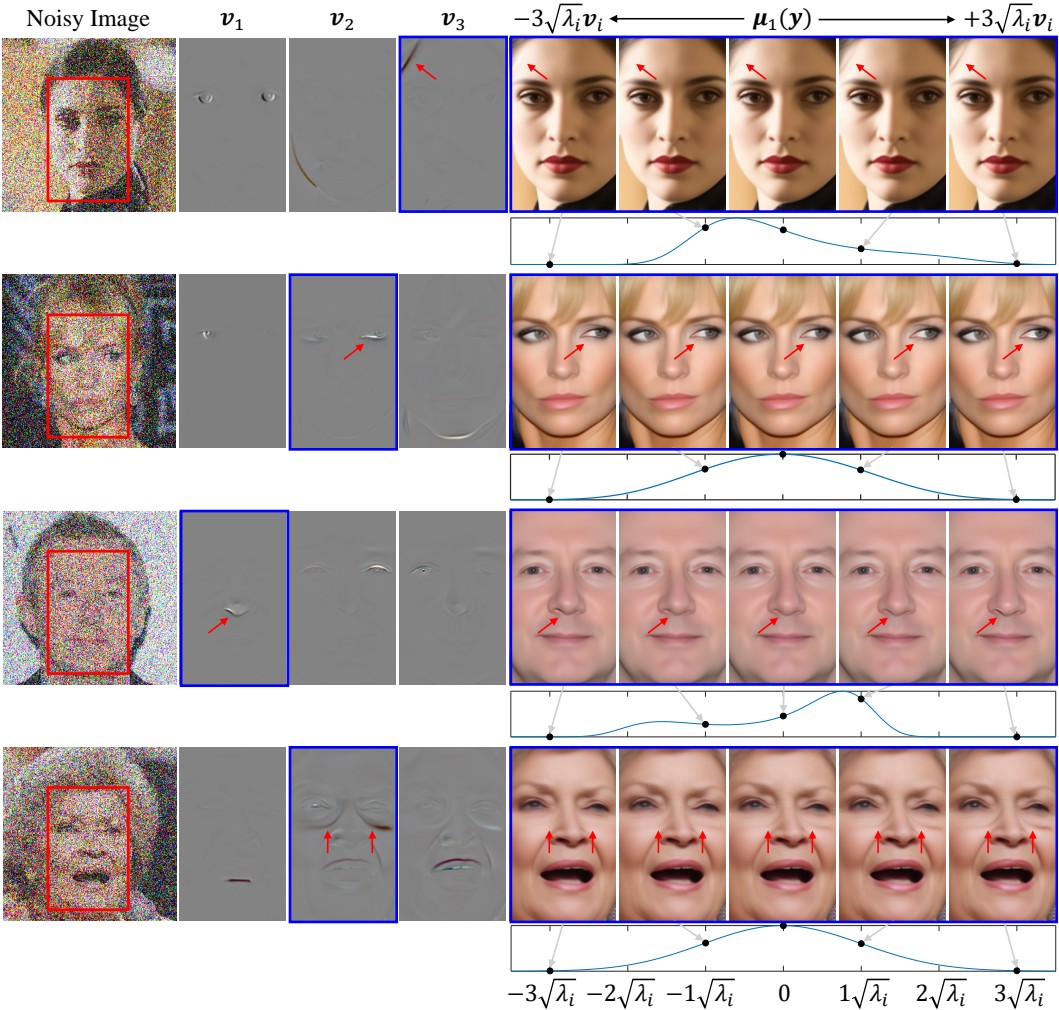

Figure S7: **Additional examples on face images, using a polynomial fit marginal distribution estimate**. In each row, the first three PCs corresponding to the noisy image are shown on the left, and one is marked in blue. On the right, images along the marked PC are shown above the marginal posterior distribution estimated for this direction. The principal components highlight meaningful uncertainty, such as eyes shape or the existence of wrinkles. Note as an example in the first row how the optimal-MSE restoration is the mean of the more probable mode, depicting no hair on the forehead, and the distribution's tail, yielding the less-probable semi-translucent hair.

