# OpenReview forum: "On the Posterior Distribution in Denoising: Application to Uncertainty Quantification"
_ICLR.cc/2024/Conference — ICLR 2024 poster_

### Official Review · Reviewer_N9TK · 2023-10-31

**Soundness:** 4 excellent
**Presentation:** 4 excellent
**Contribution:** 3 good
**Rating:** 6
**Confidence:** 3

**Summary:**

A method is proposed to quantify uncertainties when solving denoising problems with data-driven pre-trained denoisers.  The high-order central moments of the target posterior distribution are related to higher-order derivatives of its mean.  For high-dimensional problems the top eigenvectors of the posterior covariance are computed to capture the main modes of variation for uncertainty visualization.  Experiments are presented to demonstrate application of the method on a number of data-sets.

**Strengths:**

For denoising problems, denoisers implicitly encode a posterior for the data on which they are trained, which is related to the denoiser through Tweedie's formula.  Tweedie's formula explicitly relates the first posterior moment to the score of the data.  Similar relations hold for higher order momements of the posterior and higher order scores.  These expressions are presented in Theorem 1 and 2, which are proved in the appendices.  For high dimensional problems (e.g. imaging), computing these terms directly is not computationally feasible.  Instead the top eigenvectors are computed by the subspace iteration method.  These eigenvectors are then used to visualize the main modes of variation in the posterior.  The method requires only a pre-training denoiser and forward evaluations of the denoiser, avoiding any need for additional training.  It is fast and memory efficient and so suitable for high-resolution images.

**Weaknesses:**

While the methodology introduced in is very nice, the uncertainty quantification visualizations presented are a little underwhelming. The MNIST example (Fig. 4) clearly highlights the merit of the approach, where different potential modes of a 9 or 4 are apparent.  However, for the experiments performed on natural images (e.g. Fig. 5), it is difficult to distinguish much variation in the images.  I had to really zoom into these images to see the changes that are highlighed by the annotated arrows.  However, I suppose the eigenvectors themselves may be a more useful visualization aid since it is more clear what structure of the image changes.  Nevertheless, the difficulting in distringuishing changes in the modified images may limit the practical application of the method proposed.

**Questions:**

Is there any way to create modified images that more clearly show uncertainties?  I suppose increasing the scaling of the eigenvalue could be considered, although this is not very well motivated.  It might also be interesing to consider adding multiple eigenvectors at once, to show the mutliple modes of variation at once, rather than just one mode at a time.

---

> ### Author Response · Authors · 2023-11-13
>
> **“While the methodology introduced in is very nice, the uncertainty quantification visualizations presented are a little underwhelming.”, “Is there any way to create modified images that more clearly show uncertainties?”**
>
> Thanks for this comment. That’s a very important point.
>
> First, a comment is in place with regards to the side-by-side visualizations in the paper. These visualizations are not optimal, as the gradual change is hard to notice over a row of similar-looking images. The changes are much more easily noticeable in the form of a video or a GIF, where the geometric features and colors change over time, and the localized change is quite visible. We encourage you to examine the *supplementary video* (in the supplemental zip file from the original submission).
>
> As you mention, another naïve approach would be indeed to increase the factor multiplying the PCs, going beyond 3 standard deviations. However, we believe that for the purpose of uncertainty visualization, it is desirable to conform to the true uncertainty. We do not want to present to the user an image, in which the changes are more clearly seen, but which is not probable given the noisy measurement.
>
> Note that as the noise level decreases, the uncertainty is inherently reduced, and the PCs depict more subtle changes which become more difficult to distinguish. The key takeaway here is that this is not a weakness of the method, but rather a fact that stems from the problem itself. Even if we had the ground-truth covariance matrix, and visualized the ground-truth uncertainty directions, the changes would be subtle under low noise-levels, simply because the uncertainty is low in such settings.
>
> In most experiments, we visualize realistic noise-levels, in which the uncertainty is mainly in the fine details. This is not an arbitrary choice. These are the popular noise levels used in denoising tasks (namely $\\sigma=\\{25,50\\}$), and as such most pre-trained models are available for those levels. The one exception to that is the face images experiment, in which we used a high noise-level of $\\sigma=122$. This was possible because there we used a denoiser that accepts the noise as a condition, and can operate under a wide range of noise levels (we took it from a a pre-trained diffusion model). Even at that high noise-level, the general face features, their shapes and colors, can still be clearly seen in the noisy images, and thus the variations captured by the PCs are between minor and medium.
>
> Note that even minor changes can prove very important in certain contexts, like medical image reconstruction. One can envision a case in which instead of the fracture on the old vase (depicted in Fig. 5), a restoration of a noisy X-ray image is presented to a radiologist to check for a fracture of a bone. Small fractures may disappear in the MSE-optimal restoration, but uncertainty visualizations of such fine details can reveal those options and their likelihoods. This can aid the radiologist in the diagnosis.
>
> **“It might also be interesting to consider adding multiple eigenvectors at once, to show the multiple modes of variation at once, rather than just one mode at a time.”**
>
> This is indeed a nice visualization, which can easily be achieved with our method. For practical applications, it is possible to have a GUI with a slider for each principal direction. The user would then be able to either change each slider individually or modify several sliders to see the combined effect of several PCs. We will add several such examples to the supplementary of the final version. Thanks!

---

> > ### Comment · Reviewer_N9TK · 2023-11-23
> >
> > Thank you very much for the comments and clarifications.  I reviewed the animations in the supplementary video, which were indeed helpful in visualizing the uncertainties better.

---

### Official Review · Reviewer_EK8x · 2023-10-31

**Soundness:** 3 good
**Presentation:** 2 fair
**Contribution:** 1 poor
**Rating:** 3
**Confidence:** 3

**Summary:**

Authors show that given a pretrained score-based model, they are able to approximate the higher-order moments of the posterior distribution of the unknown noise-free image given noisy image.

**Strengths:**

**Originality.** While authors provide theoretical results for higher-order moments, as they mention, the second order moment has already been used in the context of denoising uncertainty quantification.

**Quality and clarity.** The paper was not too hard to follow. There are a few typos and some sentences are not clear.

**Significance.** It is unclear what this approach brings to the table that was not possible with existing uncertainty quantification methods.

**Weaknesses:**

* $\sigma$^2 not defined in Eq. 4.

* This particular method for image denoising uncertainty quantification is not motivated. Why would one use this approach? Why not just write down the posterior distribution using a likelihood model and the pretrained score-based model as (log) prior and use MCMC? Why not use amortized Bayesian inference and just simply sample from the posterior distribution?

* Related to the previous point, a lot of emphasize has been given to higher-order moments of the posterior. Why do we care about computing these? Can't we compute these quantities using samples from the posterior distribution?

* How would one use this method when the noise level is unknown?

**Questions:**

See weaknesses.

---

> ### Author Response · Authors · 2023-11-13
>
> This is comment 1 out of 2.
>
> **“The second order moment has already been used in the context of denoising uncertainty quantification.”**
>
> The second order moment has indeed been used for uncertainty quantification, but in an entirely different way, which is completely impractical for images of reasonable dimensions (it was not illustrated for images larger than 32x32). As we explain in the related work section, ref. [Meng
> et al., 2021] in the paper trained a model to estimate the second-order score and used it for quantifying uncertainty. As this approach requires training over high-order tensors, their method is limited to small images. We do not require any training and are thus less limited.
>
> **“This particular method for image denoising uncertainty quantification is not motivated. Why would one use this approach? Why not just write down the posterior distribution using a likelihood model and the pretrained score-based model as (log) prior and use MCMC? Why not use amortized Bayesian inference and just simply sample from the posterior distribution?
> Related to the previous point, a lot of emphasize has been given to higher-order moments of the posterior. Why do we care about computing these? Can't we compute these quantities using samples from the posterior distribution?”**
>
> First, note that a denoiser pretrained for a specific noise level (as we use in this work) cannot be used to generate posterior samples using MCMC. This is because the denoiser corresponds to the score of the noisy image distribution, not the clean one (namely it relates to $\\log p_\\mathbf{y}$ and not to $\\log p_\\mathbf {x}$). Approaching $\\log p_\\mathbf {x}$ is possible with a denoiser that is trained conditioned on the noise level and can handle a large range of noise-levels (e.g., as in denoising score matching and in diffusion models). Our method, on the other hand, can use *any* black-box denoiser, architecture-agnostically, that can handle only the noise-level of the test images we wish to analyze.
>
> Now, it is certainly possible to resort to some method for generating posterior samples, as you suggest. But note that simply sampling from the posterior does not provide the user with quantitative tools for assessing uncertainty, and it may also require the user tedious scrolling through thousands of samples to try to visually grasp the likelihood of each possible semantic variation in the output. Yet, as you correctly note, given many samples from the posterior, it is possible to compute e.g. the principal components (PCs) of the posterior.
>
> We followed your suggestion and compared our method to this baseline (see App. K in the new manuscript). Specifically, we used DDNM [Wang et al., 2023] to generate many samples from the posterior and then performed PCA on those samples to extract the posterior PCs. As can be seen in Fig. S4, as the number of posterior samples increases, the PCs estimated using this baseline become cleaner and more similar to our PCs. However, note that even with 500 samples, the PCs of this baseline do not seem to have fully converged, and generating each sample requires 100 neural function evaluations (NFEs) with DDNM, so that generating 500 posterior samples requires 50,000 NFEs (just for analyzing the uncertainty for a single noisy input image!). In contrast, our method can faithfully extract each posterior PC with only 10 NFEs (see convergence graph in Fig. A1). Therefore, for extracting e.g. 5 PCs, our method is roughly 1000x faster than that naïve approach.
>
> We also added a numerical comparison to that baseline. Due to the heavy computations involved in generating posterior samples with DDNM, for now we conducted comparisons only with the 4 noisy images of Fig. 2, but we will include the results for a large dataset in the revised manuscript. The comparison is as follows. First, the empirical mean of the samples generated by DDNM should theoretically approximate the posterior mean, which is the MSE-optimal restoration. As we verify, this estimate is indeed very close to our denoiser’s output, and they both achieve practically the same MSE to the ground-truth images. Second, we computed the PCs from the posterior samples (after subtracting their mean). We compared those PCs to the PCs of our method by measuring the norm of the error after projecting it onto these PCs. The larger this norm, the larger the portion of the error that these PCs account for (larger is better). Mathematically, this measure is defined as $\\|\\boldsymbol{V}^T(\\boldsymbol{x} - \\mu\_1(\\boldsymbol{y}))\\|^2\_2$, where $\\boldsymbol{V}$ is a matrix containing the PCs as columns. We found that the mean of this measure for both methods is very close and ours is even slightly better: ours is 0.81 and the baseline’s is 0.63.
>
> Thank you for this suggestion.

---

> ### Author Response · Authors · 2023-11-13
>
> This is comment 2 out of 2.
>
> **“It is unclear what this approach brings to the table that was not possible with existing uncertainty quantification methods.”**
>
> As explained in the related work section, the vast majority of existing uncertainty quantification methods only provide per-pixel uncertainty estimates. The few methods that do attempt to output posterior covariances require training and can only work on small images (due to large memory footprints).
>
> We are the first to propose a practical training-free method for computing posterior principal components using a pre-trained denoiser. In addition, we are the first to propose such a method that can be applied to any region of interest, whereas previous methods cannot.
> We are also the first to propose a way to use higher-order moments for studying the full marginal distribution along one-dimensional directions.
>
> **“$\\sigma$^2 not defined in Eq. 4.”**
>
> The standard deviation of the noise, denoted by $\\sigma$, is defined for Eq. (1), where the denoising problem is first introduced. The notation remains the same across the entire paper.
>
> **“How would one use this method when the noise level is unknown?”**
>
> This is an interesting question, which we discuss in App. I.
>
> While our theory holds only for the non-blind Gaussian denoising case, we empirically show in the paper that our approach provides sensible results also for the blind-denoising setting, in our experiment on real microscopy images (the bottom of Fig. 5), where the noise model is not known. There, the noise distribution in each pixel is likely closer to Poisson-Gaussian than to Gaussian, the noise level is unknown, and it is not even clear if the noise is completely white. To handle this task, we estimated $\\sigma$ using the naïve formula provided at the end of Sec. 3.3, and it can be seen that even in those cases our moment calculations, combined with our novel posterior PCA approach, seem to reveal meaningful uncertainty.
>
> The subject of noise-level estimation, especially for AWGN, has been heavily researched, and state-of-the-art methods reach very-high precision [Chen et al., 2015; Khmag et al., 2018; Kokil & Pratap, 2021; Liu & Lin, 2012; Liu et al. 2013]. For example, when the real $\\sigma$ equals 10, the error in estimating sigma is around 0.05 (see e.g. [Chen et al., 2015]). Even our naïve estimation method mentioned above resulted in quite accurate results, but it is possible to use more advanced noise-level estimation methods to obtain a more accurate estimate for $\\sigma$ and then apply our method.
>
> As we show in the appendix, the impact of small inaccuracies in $\\sigma$ on our uncertainty estimation is quite small. Fig. S3 shows that even for large errors of 5% in $\\sigma$ (which are significantly higher than the typical 0.5% errors for good noise level estimation techniques), our predicted eigenvectors are quite similar.
>
> Our method can in fact be motivated for blind Gaussian denoising. Specifically, suppose that $\\mathbf{y}=\\mathbf{x}+\\sigma \\mathbf{n}$, where $\\mathbf{n}\\sim\\mathcal{N}(0,\\mathbf{I})$ is independent of $\\mathbf{x}$ and the noise level $\\sigma$ is a random variable. That is, the noise level is unknown to the denoiser (only the distribution of noise levels $p_\\sigma$ is known). In this case, it is easy to show using the law of total expectation that the Jacobian of the MSE-optimal denoiser is given by $\\mathbb{E}\left[\\frac{\textnormal{Cov}(\\mathbf{x}|\\mathbf{y},\\sigma^2)}{\\sigma}\middle|\\mathbf{y}=\\boldsymbol{y}\right]$. Namely, instead of $\\frac{\textnormal{Cov}(\\mathbf{x}|\\boldsymbol{y})}{\\sigma^2}$, which we had in the Gaussian setting, here the Jacobian reveals the *mean* of the posterior covariance divided by $\\sigma^2$. This matrix is a linear combination of the posterior covariances corresponding to different noise levels, so that it captures some notion of spread about the posterior mean, similarly to the regular posterior covariance that arises in the non-blind setting. Thus, intuitively, we expect that the top eigenvectors of this matrix capture meaningful uncertainty directions, similarly to the non-blind setting.
>
> We added this intuitive explanation in the revised supplemental material, thanks.

---

> > ### Comment · Reviewer_EK8x · 2023-11-21
> >
> > I thank the authors for their through responses. I still believe that traditional posterior sampling methods would give you uncertainties regarding the denoising problem. Authors mention that
> >
> > > note that a denoiser pretrained for a specific noise level (as we use in this work) cannot be used to generate posterior samples using MCMC.
> >
> > The noise level has nothing to do with the pretrained prior model. This is factually wrong. You can train a generative model $p_{\theta} (x) \approx p(x)$ and use it as a prior in a Bayesian sense. This network has nothing to do with the noise level. The estimated noise level enters the likelihood function, e.g., $\frac{1}{2\sigma^2} \left \|\| y - x \right \|\|_2^2$ in case a standard Gaussian noise $n \sim \mathrm{N} (0, \sigma^2 I)$ where $y = x + n$. See [this paper](https://arxiv.org/abs/2309.01949) as an example of using a pretrained prior model in the context of image denoising UQ where instead of MCMC a VI method is used to capture the uncertainty.
> >
> > > it may also require the user tedious scrolling through thousands of samples to try to visually grasp the likelihood of each possible semantic variation in the output.
> >
> > Well, one could compute the pointwise standard deviation and obtain a reasonable measure of pixel-wise uncertainty (e.g., see Figure 5 in [this paper](https://openreview.net/pdf?id=P9m1sMaNQ8T))
> >
> > Since I am still not convinced why one should use this method for image denoising UQ, I retain my score.

---

> ### Author Response · Authors · 2023-11-21
>
> Thanks for the response.
> We’d like to clarify a point regarding our previous answer, which seems to have not been clear. Our response regarding a denoiser pretrained for a single noise level referred to the comment
> > Why not just write down the posterior distribution using a likelihood model and **the** pretrained score-based model
>
> with emphasis on the word “the”. We just wanted to clarify in our answer that we **didn’t** use a score-based model in our paper. A score-based model, like the one used in [this paper]( https://arxiv.org/abs/2309.01949), which you provide as an example, is driven by a denoiser trained for a wide range of noise levels. Indeed, the score $s\_\\theta(x,t)$ in their Eq. (5) corresponds to $\\nabla\_x \\log p\_t(x)$, which in turn is a linear function of the posterior mean for the noise level at timestep $t$ of the diffusion. In other words, the score-based model employs an MSE-optimal denoiser for a wide range of noise levels, one for each timestep $t$. We wanted to emphasize that in our paper, we don’t rely on such a model, but rather on a denoiser trained for a single noise level. We’re not saying that it’s impossible to use this approach (like we wrote, we illustrate this approach in App. K). We just wanted to clarify that we didn’t in our method.
>
> Regarding computing the per-pixel standard deviation using many samples from a posterior-sampler:
>
> First, as we mentioned in our previous response, we added to App. K experiments with the state-of-the-art posterior sampler DDNM, which is based on a pretrained diffusion model. As we explain, this sampler (like other diffusion-based samplers), requires 100 calls to the score model (namely 100 neural function evaluations) just to generate a single sample. Therefore, to generate e.g. 500 posterior samples for accumulating sufficient statistics for constructing per-pixel STD maps or per-pixel confidence intervals, requires 50,000 NFEs! Our approach, on the other hand, is orders of magnitude faster. This in itself is a big advantage.
>
> Second, and more importantly, many recent works (see references below) illustrated that per-pixel STD maps are limited in their ability to communicate uncertainty. If the per-pixel STD is high in a certain region of the image, this does not indicate whether the uncertainty is in the global brightness of the entire region, or in each pixel changing differently (or because of any interesting structure in between those two extremes). Therefore, this leads to inflated uncertainty estimates. This is the reason that the UQ community has recently shifted focus from per-pixel uncertainties to more spatially correlated notions of uncertainty. See for example, Fig. 1 in ([Belhasin et al., 23](https://openreview.net/pdf?id=ijSTOcngKs)) and Fig. 1 in ([Nehme et al., 23](https://openreview.net/pdf?id=nZ0jnXizyR)), as well as the uncertainty visualizations in ([Monterio et al., 20](https://arxiv.org/pdf/2006.06015.pdf)), ([Sankaranarayanan et a., 22](https://arxiv.org/pdf/2207.10074.pdf)), ([Kutiel et al. 23](https://openreview.net/pdf?id=J4QatK02Qc)) and their explanations about the limitations of per-pixel uncertainty measures.

---

### Official Review · Reviewer_a3nf · 2023-11-01

**Soundness:** 2 fair
**Presentation:** 3 good
**Contribution:** 3 good
**Rating:** 6
**Confidence:** 3

**Summary:**

Motivated by the recent success of using deep denoisers as priors (e.g, PnP and score matching),  this paper conduct connections between posterior central moments and the derivatives of posterior mean in a g higher order and recursive manor within the context of AWGN denoising. Then such conduction is applied for denoising  uncertainty visualization on multiple image datasets.

**Strengths:**

1)	The paper is well organized and easy to follow. The demonstration proof about posterior central moments and directional projection in a recursion manner is well presented and informative.

2)	The intuitive of using higher order moments to approximate posterior distribution is overall interesting.

3)	The linear approximation in Eq.15 seems to be effective to handle differentiation complexity for higher-order tensor.

4)	Finally, the paper has shown its potential for uncertainty visualization within the context of image Gaussian denoising.

**Weaknesses:**

1), This paper only considers AWGN removal, making its practical relevance to real-world denoising (e.g., Poisson, Laplacian, Speckle, etc) very limited. Moreover, it would be more interesting to show how this method can be applied to other imaging inverse problems like inpainting and super-resolution both empirically and theoretically.

2), At the same time, since the analysis is built on Gaussian distribution,  it is difficult to evaluate its theoretical contributions since the recurrence relation for the central moments of the Normal Distribution is somehow standard.

3), In this paper, no baseline methods are compared against with. No quantitative results about the uncertainty calibration such as expected calibration error etc. are reported.

**Questions:**

1), While the authors claim the ability to compute the principal components (PCs) of the posterior distribution for any specified image region, the implementation details remain unclear to me. Seems there are no clear connection between the PCs and image features. At least it is not controllable.

2), Figure 4, and Figure 5, the authors claims that the PCs show the uncertainty along meaningful directions. However, interpreting these findings remains challenging. Given that images often follow complex distributions, it's not evident how the curves presented by the authors establish a direct connection with changes in ground-truth geometric features.

3), The computational complexity should be reported in the revision.

4), More importantly, absolute error to the ground-truth should be also presented in Fig. 4 and Fig.5 to show that the uncertainty indeed can better reflect the restoration error.

---

> ### Author Response · Authors · 2023-11-13
>
> This is comment 1 out of 3.
>
> **“This paper only considers AWGN removal, making its practical relevance to real-world denoising (e.g., Poisson, Laplacian, Speckle, etc) very limited. Moreover, it would be more interesting to show how this method can be applied to other imaging inverse problems like inpainting and super-resolution both empirically and theoretically.”**
>
> Thanks for this important point. First, we would like to comment that while the theory we developed indeed holds only for the non-blind Gaussian denoising case, we empirically show in the paper that our approach provides sensible results also in more generic denoising settings. For example, the bottom part of Fig. 5 illustrates the use of our approach on real microscopy images, in which the noise model is not known. In this setting, the noise distribution in each pixel is likely close to Poisson-Gaussian rather than Gaussian, the noise level is unknown, and it is not even clear if the noise is completely white. As can be seen, even in those cases our moment calculations, combined with our novel posterior PC approach, seem to reveal meaningful uncertainty.
>
> Second, we would like to point out that our method can be motivated for slightly more general settings than just nonblind Gaussian denoising. In particular, it also makes a lot of sense for *blind* Gaussian denoising. Specifically, suppose that $\\mathbf{y}=\\mathbf{x}+\\sigma \\mathbf{n}$, where $\\mathbf{n}\\sim\\mathcal{N}(0,\\mathbf{I})$ is independent of $\\mathbf{x}$ and the noise level $\\sigma$ is a random variable. That is, the noise level is unknown to the denoiser (only the distribution of noise levels $p_\\sigma$ is known). In this case, it is easy to show using the law of total expectation that the Jacobian of the MSE-optimal denoiser is given by $\\mathbb{E}\left[\\frac{\textnormal{Cov}(\\mathbf{x}|\\mathbf{y},\\sigma^2)}{\\sigma}\middle|\\mathbf{y}=\\boldsymbol{y}\right]$. Namely, instead of $\\frac{\textnormal{Cov}(\\mathbf{x}|\\boldsymbol{y})}{\\sigma^2}$, which we had in the nonblind setting, here the Jacobian reveals the *mean* of the posterior covariance divided by $\\sigma^2$, where the mean is taken over all possible noise levels $\\sigma$. This matrix is a linear combination of the posterior covariances corresponding to different noise levels, so that it captures some notion of spread about the posterior mean, similarly to the regular posterior covariance that arises in the non-blind setting. Thus, intuitively, we expect that the top eigenvectors of this matrix capture meaningful uncertainty directions, similarly to the non-blind setting. We added this explanation in the revised supplemental material.
>
> For general inverse problems, it is unlikely that there exists a closed-form expression allowing to extract the posterior covariance from a pre-trained MSE-optimal model. Perhaps there are convenient expressions for certain specific settings, but we leave this to future work. One option to address general inverse problems is via training-based solutions, however here we wanted to focus on revealing the information that is encoded within the weights of a *pre-trained* model. Another approach could be to use diffusion-based posterior sampling techniques (e.g. DDNM[Wang et al., 2023]) to generate many samples from the posterior and then perform PCA on those samples to extract the posterior PCs. However, note that the computational cost of such a technique would be very high, as each sample requires many neural function evaluations (NFEs) to generate (e.g. 100 in DDNM) and at least 500 samples are needed for obtaining accurate PCs (see the illustration we added in App. K). In contrast, our method can faithfully extract each posterior PC with only 10 NFEs (see convergence graph in Fig. A1). Therefore, for extracting e.g. 5 PCs, our method is roughly 1000x faster than that naïve approach. We added this discussion to App. J.
>
> **“At the same time, since the analysis is built on Gaussian distribution, it is difficult to evaluate its theoretical contributions since the recurrence relation for the central moments of the Normal Distribution is somehow standard.”**
>
> Please note that the density for which we derive our recurrence relation is not Gaussian. Specifically, our recurrence relation is for the posterior $p_{\\mathbf{x}|\\mathbf{y}}(\\boldsymbol{x}|\\boldsymbol{y})$ corresponding to the observation model $\\mathbf{y}=\\mathbf{x}+\\mathbf{n}$. Here, $\\mathbf{n}$ is Gaussian but $\\mathbf{x}$ is not necessarily Gaussian (it can have an arbitrary distribution), so that $p_{\\mathbf{x}|\\mathbf{y}}(\\boldsymbol{x}|\\boldsymbol{y})$ is not necessarily Gaussian. Therefore, the known recurrence relations for the central moments of a Gaussian distribution do not apply in our setting.

---

> ### Author Response · Authors · 2023-11-13
>
> This is comment 2 out of 3.
>
> **“In this paper, no baseline methods are compared against with. No quantitative results about the uncertainty calibration such as expected calibration error etc. are reported.”, “Absolute error to the ground-truth should be also presented in Fig. 4 and Fig.5 to show that the uncertainty indeed can better reflect the restoration error.”**
>
> Thanks for this important point. Please note that it is impossible to *directly* measure the quality of the estimated posterior PCs because denoising datasets contain only one clean image $\\boldsymbol{x}$ for each noisy image $\\boldsymbol{y}$. This single $\\boldsymbol{x}$ is just one sample from the posterior $p\_{\\mathbf{x}|\\mathbf{y}}$ and therefore it cannot be used to extract a “ground-truth” posterior covariance matrix or “ground-truth” PCs.
>
> However, it is possible to compare our approach to a very strong baseline - that of diffusion-based posterior samplers, as mentioned above. We added this comparison to App. K. For each noisy image, we generated many posterior samples using DDNM[Wang et al., 2023] and used them to calculate the PCs of the posterior. As can be seen in Fig. S4, as the number of posterior samples increases, the PCs estimated using this baseline become cleaner and more similar to our PCs. However, note that even with 500 samples, the PCs of this baseline do not seem to have fully converged, and generating 500 posterior samples using DDNM requires 50,000 NFEs (just for analyzing the uncertainty for a single noisy input image!).
>
> We also added a numerical comparison. Due to the heavy computations involved in generating posterior samples with DDNM, for now we conducted comparisons only with the 4 noisy images of Fig. 2, but we will include the results for a large dataset in the revised manuscript. The comparison is as follows. First, the empirical mean of the samples generated by DDNM should theoretically approximate the posterior mean, which is the MSE-optimal restoration. As we verify, this estimate is indeed very close to our denoiser’s output, and they both achieve practically the same MSE to the ground-truth images. Second, we computed the PCs from the posterior samples (after subtracting their mean). We compared those PCs to the PCs of our method by measuring the norm of the error after projecting it onto these PCs. The larger this norm, the larger the portion of the error that these PCs account for (larger is better). Mathematically, this measure is defined as $\\|\\boldsymbol{V}^T(\\boldsymbol{x} - \\mu\_1(\\boldsymbol{y}))\\|^2\_2$, where $\\boldsymbol{V}$ is a matrix containing the PCs as columns. We found that the mean of this measure for both methods is very close and ours is even slightly better: ours is 0.81 and the baseline’s is 0.63.
>
> Finally, we verified the predicted eigenvalues by comparing the projected test error over the 1st PC, $\\boldsymbol{v_1}^T(\\boldsymbol{x} - \\mu_1(\\boldsymbol{y}))$, to the predicted 1st eigenvalue $\lambda_1$. The average ratio between those two quantities should theoretically be 1. For the 17 noisy images appearing in the manuscript, we found that the average of this ratio is 0.94. In the final version we will expand this experiment to a larger set of noisy test images.
>
> **”The computational complexity should be reported in the revision.”**
>
> Our computational complexity is only dependent on the chosen number of subspace iterations $K$ and requires $K+1$ NFEs. In all experiments we used $K=50$, so that our method involved 51 NFEs per input image. Yet, as can be seen in Fig. S1, convergence is typically achieved after around 10 iterations, so that in principle the method could be run with only 11 NFEs.
>
> **“While the authors claim the ability to compute the principal components (PCs) of the posterior distribution for any specified image region, the implementation details remain unclear to me. Seems there are no clear connection between the PCs and image features. At least it is not controllable.”**
>
> First, we would like to point out that our code is available in the provided anonymous repository, which might help mitigate any unclarity implementation-wise.
>
> $\\boldsymbol{v}$ can be masked during Alg. 1, which makes sure that the computed PC affects only the desired region (set by the mask). This can be seen in the different widths and heights of the various PCs displayed in the experiments. This is an important point which we agree was not described clearly in the paper and will fix it in the final version. Thank you for pointing it out.

---

> ### Author Response · Authors · 2023-11-13
>
> This is comment 3 out of 3.
>
> **“Figure 4, and Figure 5, the authors claims that the PCs show the uncertainty along meaningful directions. However, interpreting these findings remains challenging. Given that images often follow complex distributions, it's not evident how the curves presented by the authors establish a direct connection with changes in ground-truth geometric features.”**
>
> That’s a good point. To convince that the distributions we estimate along the PCs are accurate, we measured the negative log likelihood (NLL) of the GT images projected onto those directions (lower is better). We compared this to the NLL of a Gaussian distribution defined by only the first two moments. For the images of Fig. 4, Fig. 5, Fig. S5, Fig. S6 and Fig. S7, the average NLL of these distributions is 4.63 and the average NLL of the Gaussian distributions is 5.75. For the final manuscript, we will include an evaluation over a large test set.

---

> ### Comment · Reviewer_a3nf · 2023-11-22
> **Thank You for Addressing My Comments**
>
> The reviewers would like to thank the authors for taking the time to address the reviewers' comments. I indeed enjoy reading this manuscript. Please including the additional results into the final version. The connections between the PCs and image features and its implementation need to be clearly discussed in the supplements at least. Finally, I will increase my rating to reflect that this is an interesting paper to read.

---

### Official Review · Reviewer_HsMN · 2023-11-02

**Soundness:** 4 excellent
**Presentation:** 4 excellent
**Contribution:** 3 good
**Rating:** 8
**Confidence:** 5

**Summary:**

This paper proposes a new method to compute posterior central moments of minimum mean squared error (MMSE) estimates from observations contaminated by additive white Gaussian noise. Then, this theoretical development is used to obtain uncertainty estimates in the context of image denoising. The method is cleverly adapted to the case where the denoiser is a black box (e.g., a neural network) only requiring forward passes through it, rather than back-propagation to compute derivatives exactly.

**Strengths:**

The method proposed in the paper is a relevant contribution to the important problem of estimating uncertainty in high-dimensional estimation problems, in this case, in image denoising from additive white Gaussian noise. The paper is very well written, with very precise and clear notation, and is a pleasure to read. The graphical illustrations of the results are very clear and useful and the experiments are also well presented, although not as impactful, as the variations depicted are very subtle.

**Weaknesses:**

The paper does not have, in my opinion, any major weaknesses, although a few minor things could be improved.

When the authors first mention denoising (second line of the introduction), they cite a couple of papers, the oldest of which is from 2017. This may give the wrong idea that image denoising started in 2017, when in fact it is arguably the oldest and longest-standing problem in image processing, going back at least to the 1960s. More classical references should be mentioned here, rather than just a couple of recent deep-learning-based methods.

There are two common meanings for "score": gradient of the log-likelihood w.r.t. the parameters (more common in statistics) or w.r.t. the observations (more common in machine learning). It would be nice to make clear that you're using the second one, to make sure that some statistician reading the paper doesn't get confused.

Equation (1) is not a denoising problem; it is the observation model underlying a denoising problem.

According to Efron (2011), Tweedie's formula was derived by Robbins in 1956, which predates Miyasawa (1961).

Notice that the first equality in Equation (4) is basically equivalent to Equation (2.8) in the paper by Efron (2011).

Minor typos: "memory efficient" -> "memory-efficient"; "...which connects between the MSE-optimal denoiser and the score function of noisy signals" -> "...which connects the MSE-optimal denoiser with the score function of noisy signals"; "...the most well known ..." -> "... the best known ..." or "...the most well-known ...".

**Questions:**

I have no questions.

**Details Of Ethics Concerns:**

Not applicable.

---

> ### Author Response · Authors · 2023-11-13
>
> We thank the reviewer for the thorough and kind review.
>
> In the revised version, we incorporated all comments and suggestions.
>
> In particular, we added citations to more classical denoising papers and clarified the meaning of the term score in the introduction. We also rephrased our reference to Eq. (1), calling it an observation model rather than a denoising problem.
>
> We thank the reviewer for pointing out Robbins’ derivation from 1956, which we weren’t aware of. We added a citation to that paper.
> We also corrected all typos. Many thanks!

---

### Meta-Review · Area_Chair_W2KV · 2023-12-07

**Metareview:**

This article provides a method to approximate the posterior distribution in denoising models with additive Gaussian noise, under a general prior distribution on the signal. The approach is based on an identity, derived by the authors, that relates the central moments of the posterior distribution to the derivative of the posterior mean with respect to the data. For high-dimensional problems, the top eigenvectors of the posterior covariance are computed to capture the main modes of variation for uncertainty visualisation. The paper is very well written, and all but one reviewer acknowledged that the approach was novel and of interest. In their reviews, the reviewers raised several concerns, particularly:

1/The method is restricted to models with additive Gaussian noise.

2/The experimental section does not fully demonstrate the usefulness of this approach for uncertainty visualisation.

3/ There is a need for a better discussion comparing this approach to alternatives.

One reviewer still had concerns about the general motivation after considering the author's response. In my opinion, and in the opinion of other reviewers, the authors have satisfactorily addressed these concerns in their response and the revised version of the paper. After careful consideration, I recommend this paper to be accepted.

**Justification For Why Not Higher Score:**

This is a nice paper, but the novelty is somewhat limited.

**Justification For Why Not Lower Score:**

The paper is well written, and provides a nice method for uncertainty visualisation in denoising problems with additive Gaussian noise.

---

### Decision · Program_Chairs · 2024-01-16

Accept (poster)